# RPP: A Certified Poisoned-Sample Detection Framework for Backdoor Attacks under Dataset Imbalance

## Abstract

Deep neural networks are highly susceptible to backdoor attacks, yet most defense methods to date rely on balanced data, overlooking the pervasive class imbalance in real-world scenarios that can amplify backdoor threats. This paper presents the first in-depth investigation of how the *dataset imbalance* amplifies backdoor vulnerability, showing that (i) the imbalance induces a majority-class bias that increases susceptibility and (ii) conventional defenses degrade significantly as the imbalance grows. To address this, we propose **Randomized Probability Perturbation (RPP)**, a *certified poisoned-sample detection framework* that operates in a black-box setting using only model output probabilities. For any inspected sample, RPP determines whether the input has been backdoor-manipulated, while offering provable within-domain detectability guarantees and a probabilistic upper bound on the false positive rate. Extensive experiments on five benchmarks (MNIST, SVHN, CIFAR-10, TinyImageNet and ImageNet10) covering 10 backdoor attacks and 12 baseline defenses show that RPP achieves significantly higher detection accuracy than state-of-the-art defenses, particularly under dataset imbalance. RPP establishes a theoretical and practical foundation for defending against backdoor attacks in real-world environments with imbalanced data.

## 1 Introduction

Deep neural networks (DNNs) are increasingly embedded in safety-critical applications ranging from autonomous driving (Kong et al., 2020; Wen & Jo, 2022) to facial recognition systems (Yang et al., 2021; Anusudha et al., 2024). However, they are known to be vulnerable to backdoor attacks (Gu et al., 2019; Goldblum et al., 2022; Zeng et al., 2023; Le Roux et al., 2024). The core concept of a neural backdoor involves embedding a unique pattern, known as a trigger, into the training data. The resulting model behaves normally on clean inputs but, when the trigger appears, reliably redirects predictions to an attacker-chosen target class. For example, a small sticker on a stop sign can induce a traffic system to read "go" (Liu et al., 2018b).

One major way of defending against backdoor attack, called prior-training defense, is to detect and remove poisoned samples from the dataset before the training process, evidenced by a number of defenses (Tran et al., 2018; Chen et al., 2021; Qi et al., 2023b; Pan et al., 2023; Guo et al., 2023; Pal et al., 2024). However, existing methods typically assume an idealized scenario where defenses operate on balanced training sets (or their derived models)—a condition rarely found in real-world domains. In practice, data are often highly imbalanced, with some classes far more prevalent than others (He & Garcia, 2009; Fernández et al., 2018; Johnson & Khoshgoftaar, 2019; Aguiar et al., 2024). For example, in medical diagnosis datasets, healthy samples might outnumber cancer cases by more than 1000:1. Likewise, in autonomous driving, minority classes (e.g., rare traffic signs or road anomalies) are inherently underrepresented. In Sec. 3, we conducted a comprehensive investigation that reveals two key phenomena in this imbalance context: (1) We identify a significant correlation between the efficacy of backdoor attacks and the degree of dataset imbalance, demonstrating that imbalanced datasets are inherently more susceptible to backdoor attacks. (2) We further observe that the performance of existing backdoor defense mechanisms degrades significantly as data imbalance intensifies. On the other hand, existing approaches lack formal theoretical guarantees and remain susceptible to adaptive adversaries (Athalye et al., 2018; Liu et al., 2022; Zhong et al., 2025), thereby

perpetuating a cat-and-mouse dynamic between attackers and defenders, which is not affordable for safety-critical applications. *Therefore, there is an urgent need for a certified prior-training defense that is resilient to imbalanced data.*

To this end, we propose *Randomized Probability Perturbation (RPP)*, **a certified prior-training poisoned sample detection that maintains its robustness across both balanced and imbalanced data.** Unlike existing defenses that rely on distribution-level signals (such as clustering or class-level metrics to separate benign and poisoned samples) which often fail with imbalanced data, RPP examines individual sample behaviors under noise, making it resilient to distribution shifts. Specifically, we observe that poisoned inputs exhibit distinctive, consistent patterns in their prediction probabilities as random perturbations are applied—patterns that are noticeably different from those of benign samples. By quantifying the per-input stability of the predictive probability vector and calibrating detection thresholds via a conformal prediction framework, RPP identifies suspicious samples even when they are rare in imbalanced datasets. This *sample-level* perspective thus bypasses the heavy reliance on uniform or abundant class representations, making RPP naturally suited to both balanced and imbalanced scenarios. Moreover, Sec. 5 derives a certified condition under which backdoored samples are guaranteed to be detectable, yielding provable performance guarantees and enabling practical deployment—especially under severe class imbalance.

Our **contributions** are summarized as follows: **(1)** We investigate and report a strong correlation between the degree of data imbalance and the success rate of backdoor attacks, demonstrating that highly imbalanced datasets are intrinsically more vulnerable to such threats. **(2)** We show that conventional backdoor defense mechanisms degrade significantly in performance as data imbalance intensifies, highlighting the need for new defensive approaches tailored to skewed class distributions. **(3)** Guided by these observations, we introduce RPP, a prior-training poisoned sample detection technique that adapts effectively to severe class imbalance. We further provide certified detection guarantees and a rigorous theoretical foundation, offering a robust framework for future backdoor defense strategies.

## 2 RELATED WORK

Backdoor attacks can be categorized into two primary types based on their threat models.

**(1) Model-Manipulation Backdoor.** The first type, model manipulation attacks, operates under a strong but less practical assumption where attackers can control the training process, such as Input-aware Nguyen & Tran (2020), and LIRA Doan et al. (2021). A range of *empirical* works have been proposed to detect or repair trained models to mitigate backdoor threats. Currently, common approaches include Neural Cleanse (Wang et al., 2019) and Artificial Brain Stimulation (Liu et al., 2019), which recover potential triggers to erase the backdoor, as well as fine-tuning on clean auxiliary data (Liu et al., 2017) or directly pruning backdoor-related neurons (Liu et al., 2018a). Some certified defense methods also be proposed (Wang et al., 2020a; Xie et al., 2021; Xiang et al., 2024). *In this work, we do not consider such model manipulation backdoor.*

**(2) Data-Poisoning Backdoor.** In our work, we focus on the second type with a more practical assumption—data poisoning attacks—which presents a reasonable scenario where attackers have no access to the training process, but can only poison a small portion of the training data by acting as malicious data providers. This category ranges from simple pixel or blend triggers (e.g., BadNets (Gu et al., 2019), Blend (Chen et al., 2017)) to more stealthy or geometric triggers (e.g., ISSBA (Li et al., 2021b), WaNet (Nguyen & Tran, 2021)) and clean-label strategies (e.g., CL (Turner et al., 2019), Sleeper Agent (Souri et al., 2022), SIG (Barni et al., 2019), Narcissus (Zeng et al., 2023)).

*Empirical Defenses.* The research community has reported a number of empirical defenses aimed at detecting and removing poisoned samples during the training process. These defenses span feature-outlier pruning (SS (Tran et al., 2018)), reverse-engineering optimization for training-set cleansing(RE (Xiang et al., 2021)), maximum-margin activation clipping (MMDF (Wang et al., 2024b)), unlearning to break trigger–label ties (ABL (Li et al., 2021a)), simulator-discrepancy filtering with a clean proxy (De-Pois (Chen et al., 2021)), correlation decoupling via random labels (CT (Qi et al., 2023b)), test-time entropy/perturbation screening (STRIP, BBCaL (Gao et al., 2019; Hu et al., 2024)), and topology/margin heuristics (MM-BD, TED, IBD-PSC (Wang et al., 2024a; Mo et al., 2024; Hou et al., 2024)).

*Certified Defenses.* However, these empirical approaches lack formal guarantees and are vulnerable to adaptive attacks (Athalye et al., 2018; Liu et al., 2022; Zhong et al., 2025), fueling a cat-and-mouse dynamic that is unacceptable in safety-critical settings. Inspired by certified defenses for adversarial examples (Rosenfeld et al., 2020; Levine & Feizi, 2020; Jia et al., 2022; Weber et al., 2023) that provide guarantee robustness to small input perturbations rather than backdoors, CBD (Xiang et al., 2024) is the first *certified* detector of backdoored models. In specific, CBD targets *post-training* model inspection to determine whether a trained model is compromised, rather than detecting and removing poisoned training samples *before* training. A detailed comparison between CBD and our method, RPP, is provided in App. D. *To the best of our knowledge, no prior work provides certified identification and removing of mailicous training samples in data-poisoning backdoor attacks.*

Moreover, existing countermeasures implicitly assume an idealized scenario where defenses operate on balanced training sets—an assumption rarely met in real-world domains. In practice, datasets are often highly imbalanced, with some classes far more than others (He & Garcia, 2009; Fernández et al., 2018; Johnson & Khoshgoftaar, 2019; Aguiar et al., 2024). Our threat investigation (Sec. 3) demonstrates that imbalanced datasets not only significantly increase attack risks (i.e., high attack success rates with low attack budgets) but also drastically reduce the defense performance of existing countermeasures (i.e., low detection accuracy and high false positive rates). In sharp contrast, our proposed method, RPP, is a certified poisoned sample detector that maintains its performance across both balanced and imbalanced data (Sec. 4).

# 3 THREAT INVESTIGATION

**Threat Model.** We focus on backdoor attacks in classification settings under a representative *data-poisoning* threat model, where the adversary can inject poisoned training samples but cannot manipulate the training procedure or final model (Oprea et al., 2022; Cinà et al., 2024). We further consider a practical yet challenging aspect of real-world settings: dataset imbalance. We assume that the attacker is aware of the general distribution of the victim's dataset and how the majority bias in imbalanced datasets affects the success rate of backdoor attacks. As such, the attacker strategically poisons data in minority classes by altering their labels to those of majority classes, which has proven to enhance the attack effectiveness and reduce the attack budget (Sec. 3). *In this paper, we do not consider multi-trigger or multi-target backdoors (Xue et al., 2020), as even empirical detection remains challenging (Xiang et al., 2024).*

**Formal Backdoor Attack Setting.** We consider $K$-class classification with inputs $\mathcal{X} \subset \mathbb{R}^d$ and labels $\mathcal{Y} = \{1, \ldots, K\}$. A classifier $f(\cdot \mid w)$ maps $x \in \mathcal{X}$ to a predictive distribution $\boldsymbol{p}(x \mid w) \in [0, 1]^K$. The $y$-th entry, $p_y(x \mid w)$, is defined as $p_y(x \mid w) = \mathbb{P}(f(x \mid w) = y), \forall y \in \mathcal{Y}$.

Let $\mathcal{D} = \{(x_i, y_i)\}_{i=1}^N$ be the collected dataset. In a backdoor attack, an adversary chooses a trigger $\delta$ and target class $y_t$, producing a poisoned set $\mathcal{D}(x + \delta)$ by stamping $\delta$ and assigning label $y_t$, so that $f(\cdot \mid w) = y$ for benign inputs while $f(x + \delta \mid w) = y_t$ for triggered ones. We also assume a clean calibration set $\mathcal{V} = \{(x_i, y_i)\}_{i=1}^n$ drawn i.i.d. from the same distribution as $\mathcal{D}$.

**Investigation.** Previous studies (Wang et al., 2020b; Pang et al., 2024) have shown that minority classes in long-tail distribution are particularly susceptible to data poisoning attacks. Building on these insights, we investigate whether the inherent classification bias toward majority classes further amplifies this vulnerability in different distributions: specifically, if an attacker deliberately targets underrepresented classes to be misclassified as a majority class, does the existing bias increase the likelihood of the attack's success?

To simulate imbalanced scenarios, we modify the training set in two ways: *long-tailed* (Cui et al., 2019) and *step imbalance* (Buda et al., 2018). The imbalance ratio $\rho$ quantifies the degree of imbalance, defined as the ratio between the largest and smallest class sizes: $\rho = \frac{\max_i\{n_i\}}{\min_i\{n_i\}}$. Unless otherwise specified, we consider $\rho = 2, 10, 100, 200$ (see Fig. 6). In the long-tailed setting, the sizes of the classes decay exponentially from head to tail, leading to a progressive decline across different classes. In the step imbalance setting, we assign a uniform, reduced sample size to all minority classes. We define $\mu$ as the proportion of minority classes, typically set at 0.9 (i.e., 9 out of 10 classes are designated as minority) across all experiments to balance the distribution and guarantee a thorough evaluation. More details are provided in App. B.1. In both cases, we apply Badnets (Gu et al., 2019) attacks by randomly selecting an equal number of samples from the minority classes (classes 1–9)

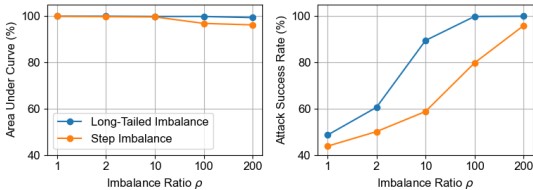

Figure 1: AUC/ASR of Badnets on MNIST with balanced ($\rho = 1$) and imbalanced ($\rho = 2, 10, 100, 200$) training sets; long-tailed ($r = 45$, $p = 0.4\%$) and step-imbalanced ($r = 18$, $p = 0.3\%$).

|  | AC | | ASSET | | SCALE-UP | |
|---|---|---|---|---|---|---|
|  | TPR | FPR | TPR | FPR | TPR | FPR |
| $\rho{=}2$ | 61.8 | 10.4 | 88.1 | 38.0 | 98.9 | 17.5 |
| $\rho{=}10$ | 57.2 | 8.9 | 97.5 | 53.1 | 100.0 | 56.8 |
| $\rho{=}100$ | 31.5 | 0.8 | 33.3 | 52.7 | 100.0 | 84.3 |
| $\rho{=}200$ | 0.1 | 0.1 | 0.0 | 0.0 | 0.1 | 0.2 |

Table 1: Comparison with AC, ASSET, and SCALE-UP on imbalanced MNIST ($\mu = 0.9$, $\rho = 2, 10, 100, 200$) against BadNets.

in each imbalanced training set (or preserving the same poisoning ratio), then relabel them as the majority class (class 0).

**Observation 1: Imbalanced datasets are more susceptible to backdoor attacks.** Fig. 1 shows that increasing the imbalance ratio $\rho$ consistently raises ASR. Even with a fixed poisoning rate $p$, imbalanced datasets are more vulnerable than balanced ones (App. C.1), corroborating the hypothesis that majority-class bias amplifies backdoor susceptibility.

**Observation 2: Existing defenses deteriorate under imbalance due to distribution-level signals.** Both empirical and certified defenses, e.g., AC (Chen et al., 2018), ASSET (Pan et al., 2023), and SCALE-UP (Guo et al., 2023), exhibit declining performance as $\rho$ grows (Tab. 1; App. C.13; App. C.14). A key reason is that many defenses rely on *distribution-level* signals, such as classwise statistics, clustering structure, or global thresholds. They become biased and unstable under class imbalance as majority classes dominate these estimates while minority classes yield noisy, underpowered signals, leading to missed triggers. Overrepresentation of majority classes thus steers models and defenses toward majority patterns, under-detecting subtle triggers in minority classes (Tab. 2). Recent work (Wang et al., 2025) adapts a backdoor defense (MMDF (Wang et al., 2024b)) to address post-training bias under data imbalance, its finding on imbalance-induced margin distortion supports our observation of degraded defense reliability.

In practical, datasets often exhibit significant imbalance (Fernández et al., 2018; Johnson & Khoshgoftaar, 2019; Aguiar et al., 2024). Hence, there is a pressing need for backdoor defenses that remain robust to such imbalance. Our proposed RPP method addresses this challenge by providing a certified poisoned samples detection using sample-level signal and calibrating detection thresholds via a conformal prediction framework, whose performance holds in both balanced and imbalanced settings.

## 4 RPP DETECTION USING SAMPLE-LEVEL SIGNAL

### 4.1 OVERVIEW OF RPP DETECTION

**Key Intuition of Sample-Level Detection under Imbalance.** Backdoor triggers are engineered to be *robust*: once present, they reliably drive a poisoned image to the target class and are largely insensitive to simple input transformations (e.g., random noise or blurring) (Chen et al., 2017; Liu et al., 2018b), keeping its predicted probability vector relatively stable. In contrast, clean images rely on naturally occurring correlations without artificially reinforced features; mild perturbations therefore induce noticeably larger changes in their probability vectors.

Motivated by the stability gap above and the limitations under imbalance, we introduce RPP—a *sample-level* robustness metric that measures the expected change in an image's probability vector under injected random noise (formalized in Sec. 4.2). Because RPP evaluates robustness *per instance* rather than from global distributional cues, it remains effective even with severe class imbalance. We further integrate RPP with a conformal prediction framework to calibrate detection thresholds in a distribution-adaptive manner, en-

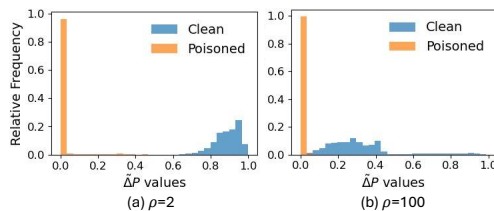

Figure 2: Relative frequency distribution of $\tilde{\Delta}P$ on the SVHN dataset with $\mu = 0.9$ under imbalance ratios: (a) $\rho = 2$ and (b) $\rho = 100$.

suring principled performance even under severe class imbalance. Empirically, poisoned samples yield consistently lower RPP than clean ones across diverse imbalance ratios; see Fig. 2 and additional results in App. C.3.

**Goal of RPP Detection.** Our objective is to determine whether a sample is backdoor-poisoned using RPP, which quantifies the expected perturbation of its predicted probability vector under random noise. Empirically (Fig. 2; App. C.3), RPP exhibits a clear separation between clean and poisoned samples across a range of imbalance ratios, enabling effective identification. Beyond empirical detection, we seek a *certified* condition guaranteeing correct identification of poisoned samples. Crucially, certification must be coupled with control of the false positive rate (FPR); thus, we explicitly target a balance between detection power and a pre-specified FPR for practical deployment.

**Outline of RPP Detection.** Suppose $\mathcal{D}(x + \delta)$ denotes the potentially poisoned dataset to be inspected, and $\mathcal{V}$ represents a clean calibration set. The detection pipeline is shown in Fig. 3. *(1) Preliminary Model Training.* First, we perform a brief training phase on the dataset to be inspected to obtain a preliminary model with acceptable performance. If the dataset contains poisoned samples, we assume that the model will have already learned the backdoor trigger (indicated by a relatively high attack success rate). *(2) Threshold Calculation.* For each $x$ in the

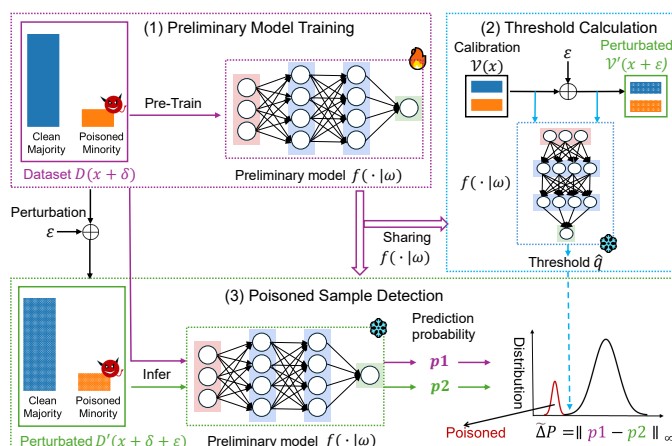

Figure 3: Overview of RPP method.

clean calibration set $\mathcal{V}$, we inject noise $J$ times, feed the perturbed inputs to the preliminary model $f(\cdot \mid w)$, and compute the empirical $\tilde{\Delta}P$ (Sec. 4.2) from the resulting probability vectors. The threshold is specified by the quantile linked to a significance level $\alpha$ over the calibration set of size $n$. *(3) Poisoned Sample Detection.* Next, we compare the empirical $\tilde{\Delta}P$ value computed for each sample in $\mathcal{D}(x + \delta)$ to the threshold. If $\tilde{\Delta}P$ is at or below this threshold, $x$ is labeled as poisoned; otherwise, it is classified as clean. The detection threshold is obtained via split conformal prediction (Aljanabi et al., 2023), providing distribution-free calibration that remains valid under class imbalance (Sec. 4.3). The step-by-step procedure is summarized as pseudocode in App. C.12.

## 4.2 DEFINITION OF RPP

Inspired by (Xiang et al., 2024), we first define the Samplewise Probability Vector (SPV), representing the prediction probability vector for each sample. Building on this, we define RPP and its empirical version based on the probability vector after noise injection.

**Definition 4.1** (Samplewise Probability Vector (SPV)). *Let $f(\cdot \mid w) : \mathcal{X} \rightarrow \mathcal{Y}$ be a pre-trained $K$-class model parameterized by $w$, where $\mathcal{Y} = \{1, \ldots, K\}$. For any input $x \in \mathcal{X}$, the SPV is the predictive probability vector $p(x \mid w) \in [0,1]^K$, whose $y$-th component $p_y(x \mid w) = \mathbb{P}(f(x \mid w) = y)$ denotes the probability that $f(x \mid w)$ assigns $x$ to class $y$. When isotropic Gaussian noise $\varepsilon \sim \mathcal{N}(0, \sigma^2 \mathbf{I})$ is added, the corresponding noisy prediction is $p_y(x + \varepsilon \mid w) = \mathbb{P}(f(x + \varepsilon \mid w) = y)$.*

**Definition 4.2** (Randomized Probability Perturbation (RPP) and empirical RPP). *Based on the SPV defined in Def. 4.1, let $\varepsilon \sim \mathcal{N}(0, \sigma^2 \mathbf{I})$ denote isotropic Gaussian noise with $\sigma > 0$. For a pre-trained model $f(\cdot \mid w)$ and input $x$, the RPP is defined as*

$$\Delta P(x \mid w, \sigma) = \mathbb{E}_{\varepsilon \sim \mathcal{N}(0, \sigma^2 \mathbf{I})} \|\boldsymbol{p}(x|w) - \boldsymbol{p}(x + \varepsilon|w)\|_\infty. \tag{1}$$

*Moreover, if $\varepsilon_j \in \mathbb{R}^d, j \in [J]$ are $J$ samples independently drawn from the Gaussian distribution $\mathcal{N}(0, \sigma^2 \mathbf{I})$, the empirical RPP (eRPP) is defined as*

$$\tilde{\Delta}P(x \mid w, \sigma) = \frac{1}{J} \sum_{j=1}^{J} \|\boldsymbol{p}(x|w) - \boldsymbol{p}(x + \varepsilon_j \mid w)\|_\infty. \tag{2}$$

**Remarks:** The RPP quantifies the bias of the probability vector by adding the randomization on the input. It is trivial that $\Delta P(x \mid w, \sigma) \to 0$ as $\sigma \to 0$ for any $x$ and $w$. In practice, we estimate the RPP by it empirical version.

### 4.3 THRESHOLD DETERMINATION OF RPP

Although the $\tilde{\Delta}P$ distributions for benign and poisoned samples are distinct (Fig. 2 and App. C.3) , determining an appropriate threshold under data imbalance setting for identifying a malicious example remains challenging. To address this challenge, we adapt conformal prediction techniques (Vovk et al., 2005; Angelopoulos et al., 2023), which are particularly well-suited for backdoor detection in imbalanced settings because they require only a single model fit and a small calibration dataset that shares a similar distribution with the test points—a requirement that aligns well with our threat model settings. The specific implementation details of our approach are described below.

Conformal prediction is traditionally used to construct prediction sets for any model (Angelopoulos et al., 2023). Given an i.i.d. calibration set $\{(x_i, y_i)\}_{i=1}^n$, it computes a score function $s(x_i, y_i)$ for each point, then selects a threshold $\hat{q}$ as the $\frac{\lceil (n+1)(1-\alpha) \rceil}{n}$ quantile of these scores. For a new test example $(x_{\text{test}}, y_{\text{test}})$, the corresponding prediction set $\mathcal{T}(x_{\text{test}})$ satisfies $\mathbb{P}(y_{\text{test}} \in \mathcal{T}(x_{\text{test}})) \geq 1 - \alpha$, where $\alpha \in [0, 1]$ is the user-specified error rate. In our approach, instead of producing prediction sets, we focus on the distribution of $\Delta P(x \mid w, \sigma)$, enabling us to directly apply conformal methodology without building class-conditioned sets. We further preserve the i.i.d. assumption by storing a small, balanced, and clean calibration dataset offline, which can be readily satisfied during the initial model deployment phase, where the calibration data is pre-loaded alongside the model; we then re-sample from this dataset—adjusting to the local data distribution—to form a calibration subset that remains contamination-free.

Under this framework, if $x_{\text{test}}$ is benign, the usual conformal properties hold; we prove in Sec. 5.2 that $\mathbb{P}(\Delta P(x_{\text{test}} \mid w, \sigma) \geq \hat{q}) \geq \min\left(1, 1 + \frac{1}{n+1} - \alpha\right)$, where $\hat{q}$ is the $\frac{\lceil \alpha(n+1) \rceil}{n}$ quantile of the calibration scores. Equivalently, if $\Delta P(x_{\text{test}} \mid w, \sigma) \leq \hat{q}$, then with high probability $x_{\text{test}}$ has been poisoned. *This makes conformal prediction well-suited for threshold selection under data imbalance, offering a principled basis for backdoor trigger detection in skewed datasets.*

**RPP for Imbalanced Dataset.** Unlike traditional methods reliant on distribution-level characteristics, RPP evaluates each sample individually based on its stability under noise perturbations. This sample-centric evaluation inherently mitigates the adverse effects of dataset imbalance. The conformal prediction framework further enhances this approach in imbalanced settings by calibrating the detection threshold using empirical RPP distributions from clean validation data. When model bias arises due to imbalance, both RPP values and the corresponding threshold adjust accordingly, preserving detection validity. This *self-normalizing property* enables RPP to adapt automatically to varying class distributions without requiring explicit adjustment. As shown in Fig. 2 and App. C.3, the threshold $\hat{q}$ of RPP distributions of clean and poisoned samples are clearly distinguishable with approximately 0.001 for imbalanced datasets with $\rho = 200$, while around 0.5 for balanced datasets.

## 5 THEORETICAL DISCUSSION

### 5.1 RPP CERTIFICATION

Beyond detection, we provide a certification that guarantees detectability of poisoned samples under specific conditions, with detailed proofs in App. A.

**Theorem 5.1.** *Let $f(\cdot \mid w) : \mathcal{X} \to \mathcal{Y}$ be the classifier with the parameter $w$. Let $y_t$ be the target class of the attacker. Define*

$$p(x) = p_{y_t}(x + \delta \mid w) \tag{3}$$

*Suppose that for any specific $x \in \mathcal{X}$ and classes $y_t \in \mathcal{Y}$, there exist $\overline{p_t} \in (0, 1)$ such that*

$$\mathbb{P}(f(x + \varepsilon \mid w) = y_t) \leq \overline{p_t}, \tag{4}$$

*where $\varepsilon \sim \mathcal{N}(0, \sigma^2 \mathbf{I})$ is an injected Gaussian noise. Let $\zeta(x, \delta)$ represent the upper bound of $\Delta P(x + \delta \mid w, \sigma)$ and suppose Assumption (A.1) is satisfied.*

*A sample attacked by a backdoor with trigger $\delta$ and the target class $y_t$ is guaranteed to be detected if*

$$\|\delta\|_2 \geq \sigma \left( \Phi^{-1}(p(x) - \zeta(x,\delta)) - \Phi^{-1}(\overline{p_t}) \right) \tag{5}$$

*where $\Phi^{-1}$ is the inverse of the standard normal cumulative distribution function.*

**Assumption (A.1)** Let $y_t$ be the target class of the attack. We assume the following inequality holds:

$$p_{y_t}(x + \delta + \varepsilon \mid w) < p_{y_t}(x + \delta \mid w) \tag{6}$$

We note that Assumption (A.1) is a reasonable and common assumption that aligns with standard practices in the field, as evidenced by its adoption in numerous defense mechanisms (Cohen et al., 2019; Xiang et al., 2024). Experiments also used to support the assumption, shown in App. C.4.

Thm. 5.1 establishes a lower bound on the trigger size necessary to detect the poisoned sample. Specifically, a sample attacked by a backdoor is guaranteed to be detected with trigger size $\|\delta\|_2 \geq R = \sigma \left( \Phi^{-1}(p(x) - \zeta(x,\delta)) - \Phi^{-1}(\overline{p_t}) \right)$. Notably, the lower bound $R$ in Thm. 5.1 is relatively small and ensures that most poisoned samples can be detected. In practice, if the trigger size is too small ($\|\delta\|_2 \leq R$), the attack success rate remains low under normal settings and attack budgets (see App. C.2).

We provide an upper bound of the trigger size in Cor. 5.2. As long as the backdoor is successfully injected, it implies that $p(x) \to 1$ and $\overline{p_t} \to 0$ (note that $\overline{p_t}$ is the upper bound of $\mathbb{P}(f(x+\varepsilon \mid w) = y_t)$ in equation 4), leading to $\left( \Phi^{-1}(p(x)) - \Phi^{-1}(\overline{p_t}) \right) \to \infty$ for any given $x$ and $w$. This means that the upper bound of $\|\delta\|_2$ is an approximately infinite number.

**Corollary 5.2.** *Suppose all conditions in Thm. 5.1 are satisfied and Assumption (A.1) is satisfied. The trigger $\delta$ of a backdoor attack satisfies that*

$$\|\delta\|_2 \leq \sigma \left( \Phi^{-1}(p(x)) - \Phi^{-1}(\overline{p_t}) \right) \tag{7}$$

According to Thm. 5.1 and Cor. 5.2, we can establish the range of trigger sizes that guarantee detection. This requires estimating $p(x), \zeta(x,\delta)$, and $\overline{p_t}$. Here, $p(x)$ denotes the probability that the sample $x + \delta$ is classified into the target class. Since the upper bound $\zeta(x,\delta)$ is not directly accessible, we estimate it using the $\frac{\lceil \alpha(n+1) \rceil}{n}$-quantile of the calibration set $\mathcal{S}$. The term $\overline{p_t}$ represents the upper bound of $\mathbb{P}(f(x + \varepsilon_j \mid w) = y_t)$. We let $y_t$ denote the class that appears most frequently among $J$ noise-injected copies of $x$, where $\epsilon_1, \ldots, \epsilon_J \sim \mathcal{N}(0, \sigma^2 I)$. Specifically, we process each $x + \epsilon_j$ through the pre-trained classifier $f(\cdot \mid w)$ and count the frequency of each class across the $J$ predictions. The class with the highest frequency is selected as $y_t$ and the count of its occurrences is denoted by $n_t$. For a given $\alpha$, we compute $\overline{p_t}$ using one-sided $(1 - \alpha)$ upper confidence intervals of the binomial distribution $B(n_t, J)$, as implemented in the function UPPERCONFBOUND($n_t, J, 1-\alpha$). The complete algorithm procedure is outlined in App. C.12.

## 5.2 THEORETICAL GUARANTEES

**Conformal Calibration Guarantee:** We formally present the calibration guarantee, ensuring that a benign example exceeds the threshold with high probability.

**Theorem 5.3.** *Let $\mathcal{V} = \{(x_i, y_i)\}_{i=1}^n$ be a calibration data set that $x_i$ is benign and i.i.d. drawn from from $\mathcal{D}$. If the test data $x_{test}$ is clean and $\hat{q}$ be the $\frac{\lceil \alpha(n+1) \rceil}{n}$ quantile as*

$$\hat{q} = \inf \left\{ q : \frac{|\{i : s_i \leq q\}|}{n} \geq \frac{\lceil \alpha(n+1) \rceil}{n} \right\}. \tag{8}$$

*Then we have that*

$$\mathbb{P}\left( \Delta P(x_{test} \mid w, \sigma) \geq \hat{q} \right) \geq 1 + \frac{1}{n+1} - \alpha. \tag{9}$$

**Performance Guarantee:** The theorem establishes a performance guarantee for our proposed robust poisoned sample detection method. Specifically, Thm. 5.4 derives an upper bound for the FPR and outlines its asymptotic behavior. A numerical validation of Thm. 5.4 is provided in App. C.18.

**Theorem 5.4.** *Let $\mathcal{S} = \{s_1, \ldots, s_n\}$ be the calibration set where $s_i = \Delta P(x_i \mid w, \sigma)$ for $i = 1, \ldots, n$ and $\alpha$ be a pre-selected confidence level. Then, the FPR can be bounded as the following:*

$$\mathbb{P}\Big(\Delta P(x_{test} \mid w, \sigma) \leq \hat{q} \mid \mathcal{S}\Big) \leq \alpha + \frac{1}{n+1} \tag{10}$$

*Moreover, if we denote $Z_n = \mathbb{P}\Big(\Delta P(x_{test} \mid w, \sigma) \leq \hat{q} \mid \mathcal{S}\Big)$, then for any $\xi > 0$ the following probability holds $\lim_{n\to\infty} \mathbb{P}(Z_n \leq \alpha + \xi) = 1$.*

## 6 EXPERIMENT

### 6.1 EXPERIMENT SETTING

**Imbalanced Datasets.** We convert the originally balanced MNIST, SVHN, CIFAR-10, TinyImageNet and ImageNet10 training sets into long-tailed and step-imbalanced versions based on the imbalance ratio $\rho$ and the fraction of minority classes $\mu$, while keeping the test sets unchanged (see Fig.6). A balanced calibration set of 100 i.i.d. samples is drawn from the same distribution. More details about these datasets are deferred to App. B.1. The results for MNIST are presented in the App. C.8.

**Backdoor Attacks.** For *certification* performance, we consider two well-known backdoor attacks: Badnets (Gu et al., 2019) and the Blend attack (Chen et al., 2017). In both cases, a trigger $\delta$ is added to the original image $x$ such that $\|\delta\|_2 \geq 0.8$ (i.e., $x \mapsto x + \delta$). Although other patterns could be used, our certified robustness depends primarily on the magnitude of the backdoor perturbation and the fraction of poisoned training data, so these patterns suffice to illustrate our approach. Meanwhile, to *empirically* validate the RPP method's detection efficacy, we evaluate *ten* additional types of triggers, detailed in App. B.2.

**Evaluation Metric.** Detection performance is measured by TPR (correctly detected poisoned samples) and FPR (benign samples incorrectly flagged as poisoned).

**Training.** We use PreActResNet18 (He et al., 2016) for SVHN and CIFAR10, ResNet34 (He et al., 2016) for TinyImageNet and ViT-B/16 (Dosovitskiy et al., 2021) for ImageNet10 as classification models. The results for different architectures (VGG16, DenseNet161 and EfficientNetB0) are shown in App. C.6. We ensured that the ASR in the pre-trained models remained above a certain threshold, e.g., 50% (App. C.17) and 90% (Tab. 2), confirming the effectiveness of the backdoor. All reported results are averaged over multiple independent runs to ensure statistical robustness.

### 6.2 CERTIFICATION PERFORMANCE

In Fig. 4, we present the TPR and FPR across dataset SVHN, CIFAR-10, TinyImageNet and ImageNet10 under balanced ($\rho = 1$) and various imbalance ratios ($\rho = 2, 10, 100, 200$). To ensure robustness, we apply isotropic Gaussian noise with dataset-specific standard deviations $\sigma$: 0.6 for SVHN, 1.0 for CIFAR-10, TinyImageNet and ImageNet10. The certification process employs two different values for the user-defined error rate parameter $\alpha$: 0.05 and 0.1. For each input sample, we generate independent Gaussian noise samples ($J = 3$) to effectively measure classification probability variations while maintaining computational efficiency, more results using different $J$ are shown in App. C.5.

Our certification method demonstrates strong effectiveness across all three datasets under various class balance conditions. When $\alpha = 0.05$, for SVHN, RPP achieves certification coverage ranges from 98.8% to 100.0% across all balance scenarios, with FPRs between 1.2% and 23.9%. The CIFAR-10 dataset shows similarly robust results, with certification rates ranging from 90.4% to 99.9% and FPRs between 5.9% and 29.6%. For TinyImageNet, RPP certifies up from 99.3% to 100.0% and FPRs between 2.0% and 31.2%. Even in ImageNet10, the method achieves certification from 84.5% to 96.7%. The corresponding FPR remains relatively low, ranging from 11.6% to 24.6% across different balance ratios. All these optimal results were achieved with $\alpha = 0.05$. When increasing $\alpha$ to 0.1, we observed that, while the TPR remains largely stable, the FPR increases. Based on this trade-off analysis, we selected $\alpha = 0.05$ as the optimal parameter for our subsequent experiments. Additionally, we demonstrate the performance of the RPP across varying trigger sizes, see App. C.17.

Figure 4: Performance of RPP against BadNets attacks with perturbation magnitude $\|\delta\|_2 \geq 0.8$, measured by TPR and FPR across balanced ($\rho = 1$) and imbalanced ($\rho = 2, 10, 100, 200$) settings, with $n = 100$ and varying $\alpha$ on SVHN, CIFAR-10, TinyImageNet and ImageNet10.

| | Defenses | Badnets | | Blend | | Trojan | | SIG | | ISSBA | | WaNet | | Sleeper Agent | | AdaPatch | | AdaBlend | | Narci. | |
|---|---|---|---|---|---|---|---|---|---|---|---|---|---|---|---|---|---|---|---|---|---|
| | | TPR | FPR | TPR | FPR | TPR | FPR | TPR | FPR | TPR | FPR | TPR | FPR | TPR | FPR | TPR | FPR | TPR | FPR | TPR | FPR |
| | SS | 55.6 | 1.2 | 57.9 | 9.6 | 54.7 | 2.0 | 62.1 | 25.2 | 39.5 | 1.9 | 60.0 | 8.2 | 60.0 | 2.3 | 11.7 | 2.8 | 16.1 | 3.9 | 8.3 | 7.7 |
| | AC | 56.1 | 16.1 | 42.7 | 13.5 | 24.0 | 49.6 | 66.7 | 16.3 | 44.1 | 9.6 | 58.3 | 6.7 | 32.1 | 20.2 | 20.7 | 17.3 | 21.3 | 12.6 | 3.6 | 21.6 |
| | RE | 73.9 | 17.0 | 72.7 | 20.5 | 56.5 | 16.9 | 38.7 | 11.0 | 41.5 | 16.4 | 50.4 | 12.3 | 52.9 | 11.7 | 57.8 | 17.5 | 50.4 | 20.8 | 10.1 | 12.3 |
| | ABL | 23.6 | 2.6 | 36.2 | 8.0 | 15.2 | 0.3 | 10.9 | 3.2 | 33.7 | 0.6 | 60.3 | 11.5 | 4.9 | 1.3 | 12.1 | 0.6 | 10.3 | 4.7 | 0.0 | 0.5 |
| | CT | 75.1 | 12.6 | 84.3 | 10.2 | 82.1 | 3.2 | 90.1 | 37.6 | 91.6 | 18.3 | 92.5 | 15.7 | 75.3 | 10.4 | 68.7 | 0.6 | 77.2 | 8.5 | 10.6 | 16.3 |
| $\rho = 2$ | ASSET | 53.3 | 31.4 | 55.1 | 27.6 | 71.9 | 37.1 | 62.2 | 36.4 | 80.4 | 15.6 | 81.1 | 16.4 | 30.5 | 29.4 | 32.9 | 36.9 | 65.9 | 22.8 | 89.0 | 6.2 |
| | SCALE-UP | 95.8 | 44.0 | 75.7 | 28.4 | 55.7 | 38.9 | 99.0 | 40.1 | 90.6 | 29.8 | 85.4 | 21.1 | 32.6 | 4.6 | 76.7 | 69.2 | 72.4 | 39.7 | 60.5 | 17.2 |
| | MSPC | 88.2 | 45.1 | 90.7 | 20.4 | 59.8 | 25.9 | 80.1 | 38.9 | 78.3 | 18.8 | 89.8 | 9.6 | 67.6 | 33.8 | 64.3 | 34.8 | 70.3 | 27.8 | 77.5 | 27.1 |
| | BBCaL | 95.7 | 18.4 | 94.2 | 19.3 | 72.8 | 13.0 | 89.4 | 19.6 | 93.1 | 17.7 | 81.8 | 20.3 | 80.4 | 17.1 | 70.1 | 24.4 | 67.3 | 20.5 | 84.0 | 16.1 |
| | STRIP | 90.2 | 18.9 | 77.8 | 26.3 | 20.7 | 10.5 | 88.6 | 30.7 | 40.2 | 22.8 | 10.5 | 33.2 | 17.6 | 10.3 | 85.4 | 24.0 | 80.9 | 26.3 | 0.0 | 0.8 |
| | TED | 96.2 | 4.8 | 90.9 | 8.2 | 76.5 | 12.1 | 88.0 | 12.9 | 90.3 | 18.4 | 88.4 | 11.9 | 89.9 | 13.3 | 86.0 | 13.6 | 83.9 | 14.0 | 90.3 | 14.0 |
| | IBD-PSC | 95.3 | 6.8 | 93.6 | 9.5 | 90.3 | 10.0 | 85.2 | 12.6 | 90.3 | 13.6 | 88.5 | 12.1 | 64.3 | 14.4 | 85.3 | 9.7 | 85.6 | 14.9 | 90.5 | 22.9 |
| | Ours | 96.7 | 3.2 | 89.5 | 9.3 | 89.9 | 12.9 | 93.0 | 4.7 | 89.0 | 10.9 | 90.6 | 8.8 | 93.8 | 24.7 | 87.4 | 4.9 | 86.7 | 11.2 | 95.6 | 16.9 |
| | SS | 29.7 | 5.3 | 29.5 | 6.7 | 58.3 | 4.2 | 10.0 | 5.0 | 20.8 | 18.7 | 30.6 | 5.5 | 33.6 | 20.5 | 6.7 | 3.2 | 20.3 | 5.2 | 0.0 | 1.2 |
| | AC | 39.2 | 0.4 | 21.2 | 2.9 | 0.0 | 0.1 | 0.0 | 0.2 | 30.0 | 7.9 | 32.5 | 6.8 | 10.3 | 0.3 | 4.6 | 0.1 | 10.5 | 4.6 | 0.0 | 0.0 |
| | RE | 13.6 | 22.5 | 10.4 | 14.3 | 20.2 | 11.0 | 14.4 | 15.9 | 16.4 | 15.9 | 17.0 | 9.4 | 10.8 | 8.1 | 9.7 | 2.8 | 8.9 | 8.4 | 0.0 | 3.1 |
| | ABL | 0.0 | 0.2 | 0.0 | 0.2 | 0.0 | 0.0 | 0.0 | 0.0 | 0.0 | 2.1 | 10.2 | 1.6 | 0.0 | 1.5 | 0.0 | 1.8 | 0.0 | 1.6 | | |
| | CT | 0.0 | 14.0 | 0.0 | 4.4 | 0.0 | 2.4 | 0.0 | 0.1 | 6.1 | 7.3 | 21.6 | 5.2 | 0.0 | 12.5 | 0.0 | 0.0 | 9.3 | 9.3 | 0.0 | 9.3 |
| $\rho = 100$ | ASSET | 0.0 | 0.0 | 12.1 | 4.7 | 0.0 | 0.0 | 0.0 | 8.7 | 10.1 | 6.4 | 13.3 | 5.8 | 0.0 | 4.2 | 21.9 | 22.8 | 17.6 | 6.4 | 5.2 | 2.2 |
| | SCALE-UP | 90.7 | 45.3 | 52.0 | 41.8 | 52.1 | 20.2 | 48.9 | 10.8 | 63.7 | 32.9 | 50.1 | 24.6 | 76.3 | 42.1 | 47.1 | 22.6 | 33.3 | 30.5 | 19.8 | 17.4 |
| | MSPC | 23.4 | 10.6 | 36.1 | 9.5 | 31.4 | 11.3 | 40.8 | 9.4 | 21.6 | 3.3 | 50.2 | 16.7 | 21.5 | 0.8 | 19.8 | 0.9 | 22.7 | 6.4 | 41.1 | 28.6 |
| | BBCaL | 67.6 | 29.9 | 68.1 | 30.6 | 50.7 | 19.9 | 52.0 | 30.7 | 58.4 | 28.9 | 46.8 | 31.3 | 60.3 | 22.0 | 39.8 | 30.1 | 37.2 | 28.4 | 55.7 | 21.3 |
| | STRIP | 49.5 | 22.4 | 37.6 | 29.1 | 12.0 | 17.4 | 42.2 | 31.1 | 18.2 | 20.6 | 7.9 | 16.2 | 8.2 | 10.4 | 39.9 | 32.2 | 27.6 | 29.9 | 0.0 | 0.5 |
| | TED | 76.9 | 9.3 | 71.4 | 16.3 | 38.7 | 22.6 | 70.3 | 20.4 | 60.7 | 20.1 | 54.3 | 17.7 | 61.2 | 16.9 | 50.5 | 20.0 | 49.4 | 22.3 | 65.5 | 19.5 |
| | IBD-PSC | 85.4 | 12.0 | 74.7 | 17.5 | 47.1 | 22.9 | 69.5 | 16.7 | 58.8 | 19.5 | 63.4 | 20.3 | 44.7 | 15.8 | 58.9 | 16.6 | 63.1 | 21.4 | 61.4 | 29.7 |
| | Ours | 100.0 | 9.2 | 79.9 | 16.3 | 75.0 | 3.4 | 73.9 | 12.9 | 52.2 | 11.1 | 70.3 | 24.9 | 59.2 | 3.8 | 66.1 | 15.9 | 69.2 | 15.4 | 67.4 | 18.7 |

Table 2: Comparison with SoTA defenses on imbalanced CIFAR-10 datasets ($\mu = 0.9$, $\rho = 2, 100$). Additional results for $\rho = 1, 10, 200$ are reported in Tab. 12.

## 6.3 EMPIRICAL PERFORMANCE

Here, we present the detection performance of the RPP method against 10 types of backdoor attacks with $\alpha = 0.05$, $n=100$ and $\sigma = 1.0$. For all experiments, we ensure the dataset contains sufficient poisoned samples to allow ASR exceeding 90%. We also compare our approach with 12 empirical backdoor defense methods, as shown in Tab. 2 and Tab. 12. We found that RPP achieves comparable or even higher TPRs than the SOTA detection methods for all trigger types. However, in several instances, methods like SCALE-UP and MSPC demonstrated higher TPR, as highlighted in red in Tab. 2 and Tab. 12, albeit at the cost of significantly increased FPR. A detailed explanation on why existing methods underperform is presented in App. C.13

## 6.4 ADDITIONAL EXPERIMENT

**Impact of Noise Level.** To assess the impact of isotropic Gaussian noise standard deviation ($\sigma$) on RPP, experiments were conducted with fixed $\alpha = 0.05$ and $n = 100$. Fig. 5 illustrates the TPR and FPR across four datasets. Optimal performance requires dataset-specific $\sigma$ ranges: 0.2 to 1.2 for SVHN, 0.1 to 3.0 for CIFAR10, 0.1 to 2.0 for TinyImageNet and ImageNet10, with variability under 20%, demonstrating RPP's robustness to noise variations.

**Resistance to Potential Adaptive Attacks.** **(1) Small-norm trigger attack:** Consistent with prevailing threat models (Souri et al., 2022; Zeng et al., 2023), we employ *low* poisoning rates to preserve stealth while sustaining a high ASR. Within the theoretical regime $\|\delta\|_2 \geq 0.8$, RPP satisfies its certified detectability conditions. We further examine a worst-case *adaptive* scenario in which the adversary is defense-aware and reduces trigger strength to $\|\delta\|_2 < 0.8$; to maintain ASR, the poisoning rate is increased to $p = 10\%$ (ASR > 90%). Even under this setting, RPP frequently detects poisoned samples empirically (App. C.11). Moreover, exceedingly small triggers typically necessitate impractically high poisoning budgets, limiting their relevance in realistic deployments.

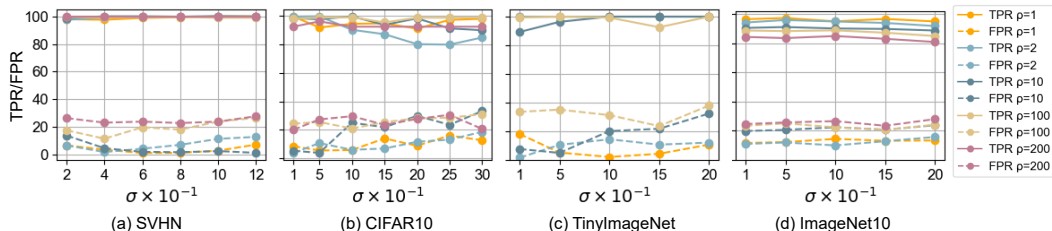

Figure 5: Performance of RPP against BadNets attacks with $\|\delta\|_2 \geq 0.8$, measured by TPR and FPR on SVHN, CIFAR-10, TinyImageNet and ImageNet10 under balanced ($\rho = 1$) and imbalanced ($\rho = 2, 10, 100, 200$) settings for varying $\sigma$, with $\alpha = 0.05$ and $n = 100$.

**(2) Hybrid-label adaptive attack:** We also assess a *hybrid-label adaptive attack*, where a defense-aware adversary poisons 80% of samples via label flipping and 20% by adding triggers with Gaussian noise while keeping original labels to induce prediction ambiguity. Despite this strategy, RPP remains robust (App. C.11), as the clean-only conformal calibration effectively filters out samples with atypical confidence fluctuations, preserving strong detection performance.

**More Experiments.** In subsequent ablation studies, we systematically investigate several key factors that may affect the performance of RPP. Specifically, we vary the size of the calibration set (App. C.7) and examine the impact of using imbalanced calibration sets (App. C.10). We further explore RPP's robustness when the backdoor target label belongs to a minority or intermediate class (App. C.15), and assess its effectiveness using class-imbalance mitigation techniques such as logits adjustment (App. C.16).

## 7 CONCLUSION

In this paper, we show that imbalanced datasets are more vulnerable to backdoor attacks than balanced ones, and existing defenses—designed for balanced data—perform poorly under imbalance. To address this, we propose RPP, the first certified poisoned samples detector that quantifies changes in prediction probabilities caused by random noise. RPP not only enables accurate detection but also provides a certified condition under which backdoors are guaranteed to be detectable. Our analysis shows that if the backdoor perturbation exceeds a threshold—ensuring high ASR with minimal poisoning—RPP can reliably distinguish poisoned from clean samples. Extensive experiments on five benchmarks confirm its strong detection and certification performance, even in imbalanced scenarios.

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

APPENDICES

CONTENTS

- **Section E: Computational Complexity**

- **Section F: Broader Impact**

- **Section G: Limitations & Future Work**

- **Section H: Use of Large Language Models (LLMs)**

# A PROOF OF THEOREMS

## A.1 PROOF OF THEOREM 5.1

We restate theorem 5.1 as theorem A.1 to enhance clarity and convenience. The proof is motivated by the approach in (Cohen et al., 2019), which leverages Lemma 4. For completeness, we restate this lemma as lemma A.2 and introduce assumption A.3 (Assumption A.1) below.

**Theorem A.1** (theorem 5.1). *Let $f(\cdot \mid w) : \mathcal{X} \to \mathcal{Y}$ be the classifier with the parameter $w$. Let $y_t$ be the target class of the attacker. Define*

$$p(x) = p_{y_t}(x + \delta \mid w) \tag{11}$$

*Suppose that for any specific $x \in \mathcal{X}$ and classes $y_t \in \mathcal{Y}$, there exist*

$$\mathbb{P}(f(x + \varepsilon \mid w) = y_t) \leq \overline{p_t} \tag{12}$$

*Let $\zeta(x, \delta)$ represent the upper bound of $\Delta P(x + \delta \mid w, \sigma)$ and suppose assumption A.3 is satisfied. A sample attacked by a backdoor with a trigger $\delta$ and the target class $y_t$ is guaranteed to be detected if*

$$\|\delta\|_2 \geq \sigma\big(\Phi^{-1}(p(x) - \zeta(x, \delta)) - \Phi^{-1}(\overline{p_t})\big) \tag{13}$$

*where $\Phi^{-1}$ is the inverse of the standard normal cumulative distribution function.*

**Lemma A.2** (Lemma 4, (Cohen et al., 2019)). *Let $X \sim \mathcal{N}(x, \sigma^2 I)$ and $Y \sim \mathcal{N}(x + \delta, \sigma^2 I)$. Let $h : \mathbb{R}^d \to \{0, 1\}$ be any deterministic or random function. Then:*

1. *If $S = \{z \in \mathbb{R}^d : \delta^\top z \leq \beta\}$ for some $\beta$ and $\mathbb{P}(h(X) = 1) \geq \mathbb{P}(X \in S)$, then $\mathbb{P}(h(Y) = 1) \geq \mathbb{P}(Y \in S)$.*

2. *If $S = \{z \in \mathbb{R}^d : \delta^\top z \geq \beta\}$ for some $\beta$ and $\mathbb{P}(h(X) = 1) \leq \mathbb{P}(X \in S)$, then $\mathbb{P}(h(Y) = 1) \leq \mathbb{P}(Y \in S)$.*

**Assumption A.3.** *Let $y_t$ be the target class of the attack. We assume the following inequality holds:*

$$p_{y_t}(x + \delta + \varepsilon \mid w) < p_{y_t}(x + \delta \mid w) \tag{14}$$

*Proof of theorem A.1.* We define the following half-spaces:

$$A := \{z : \delta^\top(z - x) \geq \sigma\|\delta\|_2 \Phi^{-1}(1 - \overline{p_t})\} \tag{15}$$

The following lemma will be used in the proof, with its proof provided at the end.

**Lemma A.4.** *Let $\overline{p_t} \in (0, 1)$ is a number defined in equation 12 and $A$ are defined by equation 15. The following equality holds:*

$$\mathbb{P}(x + \varepsilon \in A) = \overline{p_t} \tag{16}$$

According to lemma A.4 and the definitions of $\overline{p_t}$, we have that

$$\mathbb{P}(f(x + \varepsilon \mid w) = y_t) \leq \mathbb{P}(x + \varepsilon \in A) \tag{17}$$

which implies the following:

$$p_{y_t}(x + \delta + \varepsilon \mid w) = \mathbb{P}(f(x + \delta + \varepsilon \mid w) = y_t) \leq \mathbb{P}(x + \delta + \varepsilon \in A) \tag{18}$$

by applying lemma A.2 with $h(z) := \mathbf{1}[f(z) = y_t]$.

Now,

$$
\begin{aligned}
\mathbb{P}(x + \delta + \varepsilon \in A) &= \mathbb{P}(\delta^\top(x + \delta + \varepsilon - x) \geq \sigma\|\delta\|_2 \Phi^{-1}(1 - \overline{p_t})) \\
&= \mathbb{P}(\delta^\top \mathcal{N}(0, \sigma^2 I) + \|\delta\|_2^2 \geq \sigma\|\delta\|_2 \Phi^{-1}(1 - \overline{p_t})) \\
&= \mathbb{P}(\sigma\|\delta\|_2 Z \geq \sigma\|\delta\|_2 \Phi^{-1}(1 - \overline{p_t}) - \|\delta\|_2^2) \\
&= \mathbb{P}\left(Z \geq \Phi^{-1}(1 - \overline{p_t}) - \frac{\|\delta\|_2}{\sigma}\right) \\
&= \Phi(\Phi^{-1}(\overline{p_t}) + \frac{\|\delta\|_2}{\sigma})
\end{aligned}
\tag{19}
$$

where $Z \sim \mathcal{N}(0, 1)$ is the standard Gaussian distribution.

For any $x + \delta \in \mathcal{Y}$, we have that

$$
\begin{aligned}
\|\boldsymbol{p}(x + \delta \mid w) - \boldsymbol{p}(x + \delta + \varepsilon \mid w)\|_\infty &= \max_{1 \leq k \leq K} |p_k(x + \delta \mid w) - p_k(x + \delta + \varepsilon \mid w)| \\
&\geq |p_{y_t}(x + \delta \mid w) - p_{y_t}(x + \delta + \varepsilon \mid w)| \\
&\stackrel{(I)}{=} p_{y_t}(x + \delta \mid w) - p_{y_t}(x + \delta + \varepsilon \mid w) \\
&\stackrel{(II)}{\geq} p(x) - \mathbb{P}(x + \delta + \varepsilon \in A) \\
&\stackrel{(III)}{\geq} p(x) - \Phi(\Phi^{-1}(\overline{p_t}) + \frac{\|\delta\|_2}{\sigma})
\end{aligned}
\tag{20}
$$

where (I) holds because of assumption A.3, (II) follows from equation 18 and (III) follows from equation 19. Taking the expectation over $\varepsilon \sim \mathcal{N}(0, \sigma^2 \mathbf{I})$, we obtain that

$$
p(x) - \Phi(\Phi^{-1}(\overline{p_t}) + \frac{\|\delta\|_2}{\sigma}) \leq \mathbb{E}_{\varepsilon \sim \mathcal{N}(0,\sigma^2 \mathbf{I})} \|\boldsymbol{p}(x + \delta \mid w) - \boldsymbol{p}(x + \delta + \varepsilon \mid w)\|_\infty \tag{21}
$$

Let $\zeta = \zeta(x, \delta)$ be the upper bound of $\Delta P(x + \delta \mid w, \sigma)$. From equation 20, we have that

$$
0 < p(x) - \Phi(\Phi^{-1}(\overline{p_t}) + \frac{\|\delta\|_2}{\sigma}) \leq \zeta(x, \delta) \tag{22}
$$

which implies that

$$
\|\delta\|_2 \geq \sigma\big(\Phi^{-1}(p(x) - \zeta(x, \delta)) - \Phi^{-1}(\overline{p_t})\big) \tag{23}
$$

$\square$

*Proof of lemma A.4.* The proof of lemma A.4 is similar to the proofs of two claims in (Cohen et al., 2019, Theorem 2). We provide them here for completeness.

Recall that $\varepsilon \sim \mathcal{N}(0, \sigma^2 \mathbf{I})$ and $A = \{z : \delta^\top (z - x) \geq \sigma \|\delta\|_2 \Phi^{-1}(1 - \overline{p_t})\}$. Then,

$$
\begin{aligned}
\mathbb{P}(x + \varepsilon \in A) &= \mathbb{P}(\delta^\top (x + \varepsilon - x) \geq \sigma \|\delta\|_2 \Phi^{-1}(1 - \overline{p_t})) \\
&= \mathbb{P}(\delta^\top \mathcal{N}(0, \sigma^2 \mathbf{I}) \geq \sigma \|\delta\|_2 \Phi^{-1}(1 - \overline{p_t})) \\
&= \mathbb{P}(\sigma \|\delta\|_2 Z \geq \sigma \|\delta\|_2 \Phi^{-1}(1 - \overline{p_t})) \\
&= \mathbb{P}(Z \geq \Phi^{-1}(1 - \overline{p_t})) \\
&= 1 - \Phi(\Phi^{-1}(1 - \overline{p_t})) = \overline{p_t}.
\end{aligned}
$$

$\square$

## A.2 PROOF OF COROLLARY 5.2

From the inequality of equation 22, we have that

$$
\Phi(\Phi^{-1}(\overline{p_t}) + \frac{\|\delta\|_2}{\sigma}) \leq p(x) \tag{24}
$$

which implies that

$$
\|\delta\|_2 \leq \sigma\big(\Phi^{-1}(p(x)) - \Phi^{-1}(\overline{p_t})\big)
$$

## A.3 PROOF OF THEOREM 5.3

Let $s_i = \Delta P(x_i \mid w, \sigma)$ for $i = 1, \ldots, n$ and $s_0 = \Delta P(x_{\text{test}} \mid w, \sigma)$. By the exchange ability of $x_i$ and $x_{\text{test}}$ (they are i.i.d.), we have that

$$
\mathbb{P}(s_0 \geq s_k) = \frac{n + 2 - k}{n + 1}, \quad \forall k \in [n],
$$

which is because $s_0$ is equally likely to fall in anywhere between $s_1, \ldots, s_n$. Hence, we can conclude that

$$
\begin{aligned}
\mathbb{P}(s_0 \geq s_{\lceil \alpha(k+1) \rceil}) &= \frac{n + 2 - \lceil \alpha(n+1) \rceil}{n + 1} \\
&\geq 1 + \frac{1}{n + 1} - \alpha.
\end{aligned}
$$

### A.4 PROOF OF THEOREM 5.4

Let $\mathcal{S} = \{s_1, \ldots, s_n\}$ be the calibration set where $s_i = \Delta P(x_i \mid w, \sigma)$ for $i = 1, \ldots, n$. Moreover, we denote $s_0 = \Delta P(x_{\text{test}} \mid w, \sigma)$. Similar to the proof of theorem 5.3, by the exchangeability of $x_i$ and $x_{\text{test}}$ (assume $x_{\text{test}}$ is benign), we have that

$$\mathbb{P}\Big(\Delta P(x_{\text{test}} \mid w, \sigma) \leq \hat{q} \mid \mathcal{S}\Big) = \mathbb{P}(s_0 \leq s_{\lceil \alpha(n+1) \rceil} \mid \mathcal{S}) = \frac{\lceil \alpha(n+1) \rceil}{n+1} \leq \alpha + \frac{1}{n+1},$$

which proves the upper bound. Next, we discuss the asymptotic property of FPR. In the classic conformal prediction, the distribution of coverage, $\mathbb{P}(y_{\text{test}} \in \mathcal{T}(x_{\text{test}}))$, has an analytic form and it was first introduced by Vladimir Vovk in (Vovk, 2012), i.e.,

$$\mathbb{P}(y_{\text{test}} \in \mathcal{T}(x_{\text{test}}) \mid \mathcal{S}) \sim \text{Beta}(n + 1 - \lfloor \alpha(n+1) \rfloor, \lfloor \alpha(n+1) \rfloor).$$

The details of the proof of this fact see (Vovk, 2012). We note that the probability of coverage is actually equal to $\mathbb{P}(s_0 \leq s_{\lceil \alpha(k+1) \rceil})$. Indeed, during the proof of theorem 5.3, we just skip to show the probability of coverage and choose the $\frac{\lfloor \alpha(n+1) \rfloor}{n}$ quantile as the threshold while the $\frac{\lfloor (1-\alpha)(n+1) \rfloor}{n}$ quantile is used in the classic conformal prediction. Therefore, we have that $\mathbb{P}\Big(\Delta P(x_{\text{test}} \mid w, \sigma) \leq \hat{q} \mid \mathcal{S}\Big) \sim \text{Beta}(n+1-\lfloor (1-\alpha)(n+1) \rfloor, \lfloor (1-\alpha)(n+1) \rfloor)$. If we denote $Z_n = \mathbb{P}(s_0 \leq s_{\lceil \alpha(k+1) \rceil} \mid \mathcal{S})$, the mean and variance of $Z_n$ are given by

$$\mathbb{E}[Z_n] = \frac{n + 1 - \lfloor (1-\alpha)(n+1) \rfloor}{n+1} = \frac{\lceil \alpha(n+1) \rceil}{n+1}$$

$$\text{Var}[Z_n] = \frac{(n + 1 - \lfloor (1-\alpha)(n+1) \rfloor)(\lfloor (1-\alpha)(n+1) \rfloor)}{(n+1)^2(n+2)}$$

From the Chebyshev's inequality, we have that for any $\xi > 0$,

$$\mathbb{P}(Z_n \geq \frac{\lceil \alpha(n+1) \rceil}{n+1} + \xi) < \mathbb{P}(|Z_n - \frac{\lceil \alpha(n+1) \rceil}{n+1}| \geq \xi) \leq \frac{1}{\xi^2} \frac{\lceil \alpha(n+1) \rceil)(\lfloor (1-\alpha)(n+1) \rfloor)}{(n+1)^2(n+2)},$$

which implies that

$$\lim_{n \to \infty} \mathbb{P}(Z_n \leq \alpha + \xi) = \lim_{n \to \infty} \mathbb{P}(Z_n \leq \frac{\lceil \alpha(n+1) \rceil}{n+1} + \xi) = 1 - \lim_{n \to \infty} \mathbb{P}(Z_n \geq \frac{\lceil \alpha(n+1) \rceil}{n+1} + \xi) = 1.$$

## B DETAILS FOR THE EXPERIMENTAL SETTING

### B.1 DETAILS FOR THE IMBALANCED DATASETS

The original MNIST, SVHN, and CIFAR-10 datasets consist of 60,000 training images and 10,000 test images (MNIST), 73,257 training images and 26,032 test images (SVHN), and 50,000 training images and 10,000 test images (CIFAR-10), with both datasets having 10 classes. For TinyImageNet, we selected 50 classes from TinyImageNet200 as a subset for our experiments, with each class containing 500 training images. For ImageNet-10, we constructed a 10-class subset of ILSVRC-2012 (ImageNet-1K) by selecting ten semantically diverse categories. We use the original per-class splits (approximately 1,300 training images and 50 validation images per class).To create imbalanced versions, we reduce the number of training samples per class while keeping the test sets unchanged.

We use two types of imbalance: long-tailed (Cui et al., 2019) and step imbalance(Buda et al., 2018). The imbalance ratio $\rho$ measures the degree of imbalance, defined as the ratio between the largest and smallest class sizes:$\rho = \frac{\max_i\{n_i\}}{\min_i\{n_i\}}$.

In our studies, we designated imbalance ratios of $\rho = 2, 10, 100, 200$, as depicted in Fig. 6. For TinyImageNet, the ratios were set at $\rho = 2, 10, 100$. In scenarios with long-tailed distributions, the sizes of the classes decrease exponentially, leading to a progressive decline across different classes. Conversely, the step imbalance method assigns a uniform, reduced sample size to all minority classes, while maintaining larger sample sizes for more frequent classes, thus establishing a distinct separation between the two groups. We define $\mu$ as the proportion of minority classes, typically set at 0.9 across all experiments to balance the distribution and guarantee a thorough evaluation, except for TinyImageNet where $\mu = 0.98$.

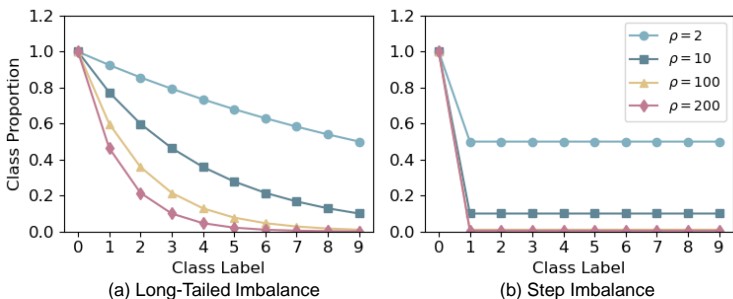

Figure 6: Class proportion types on the MNIST, SVHN, CIFAR10, TinyImageNet and ImageNet datasets. (a) Long-Tailed Imbalance ($\mu = 1/(\rho + 1)$, $\rho = 2, 10, 100, 200$). (b) Step Imbalance ($\mu = 0.9$, $\rho = 2, 10, 100, 200$).

## B.2 EXAMPLES OF POISONED SAMPLES

For **certification** performance, we consider two well-known backdoor attacks: Badnets (Gu et al., 2019) and the Blend attack (Chen et al., 2017). In both cases, a trigger $\delta$ is added to the original image $x$ such that $\|\delta\|_2 \geq 0.8$ (i.e., $x \mapsto x + \delta$). For Badnets, we use a chessboard pattern (Xiang et al., 2020) as the trigger; for the Blend attack, we adopt a "hello-kitty" image with a blending rate of 0.2. Although other patterns could be used, our certified robustness depends primarily on the magnitude of the backdoor perturbation and the fraction of poisoned training data, so these patterns suffice to illustrate our approach.

Meanwhile, to **empirically** validate the RPP method's detection efficacy, we evaluate 10 types of triggers. These include Chessboard (Xiang et al., 2020), Blend (Chen et al., 2017), Trojan (Liu et al., 2018b), Sleeper Agent (Souri et al., 2022), SIG (Barni et al., 2019),ISSBA(Li et al., 2021b)), WaNet(Nguyen & Tran, 2021), AdaPatch (Qi et al., 2023a), AdaBlend Patterns (Qi et al., 2023a) and Narcissus (Narci.) (Zeng et al., 2023), detailed in 6.3. Here, we showcase original clean samples alongside their corresponding poisoned counterparts, each embedded with one of these nine distinct triggers, as shown in Fig. 7.

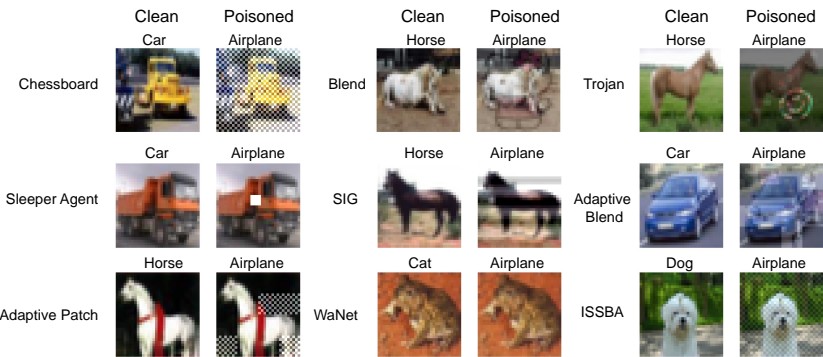

Figure 7: Examples of nine types of backdoor attacks on CIFAR10.

## B.3 DETAILS FOR THE MODEL TRAINING

We employ a standard two-layer convolutional neural network (Tab. 3) for MNIST, trained for 50 epochs with a batch size of 128 and a learning rate of $10^{-3}$ using the Adam optimizer (Kingma, 2014).

For SVHN and CIFAR-10, we train PreActResNet18 (He et al., 2016) for 100 epochs at a batch size of 128 and a learning rate of $5 \times 10^{-4}$ using Adam. The training images are augmented by random cropping and horizontal flipping.

For TinyImageNet, we train ResNet34 (He et al., 2016) for 100 epochs at a batch size of 128 and a learning rate of $5 \times 10^{-4}$ using Adam. Data augmentation includes random horizontal flipping and random cropping.

For ImageNet10, we train ViT-B/16 (Dosovitskiy et al., 2021) for 100 epochs at a batch size of 128 and a learning rate of $5 \times 10^{-4}$ using Adam. Data augmentation includes random horizontal flipping and random cropping.

| Layer | # of Channels | Filter Size | Stride | Activation |
|-------|---------------|-------------|--------|------------|
| Conv | 32 | $3 \times 3$ | 1 | ReLU |
| MaxPool | 32 | $2 \times 2$ | 2 | - |
| Conv | 64 | $3 \times 3$ | 1 | ReLU |
| MaxPool | 64 | $2 \times 2$ | 2 | - |
| FC | 128 | - | - | ReLU |
| FC | 10 | - | - | Softmax |

Table 3: Model Architecture for MNIST.

## C   MORE EXPERIMENTS

### C.1   ATTACK SUCCESS RATE WITH SAME POISONING RATIO

In Fig. 1, we observe that when an equal number of minority class samples (classes 1-9) are randomly selected and subjected to Badnets backdoor attacks (Gu et al., 2019) with their labels changed to the majority class (class 0), the ASR in imbalanced datasets is higher than in balanced datasets. In Fig. 8, we preserved the same poisoning ratio (p) across balanced ($\rho = 1$) and imbalanced training set ($\rho = 200$), and implemented the same backdoor attacks. Our findings also suggest that imbalanced datasets are more vulnerable to backdoor attacks compared to their balanced counterparts.

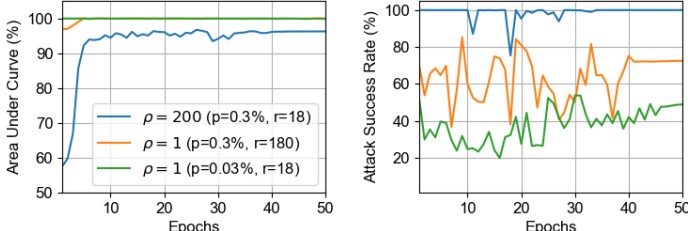

Figure 8: The AUC and ASR of the Badnets backdoor attacks on balanced ($\rho = 1$) and imbalanced ($\mu = 0.9, \rho = 200$) MNIST training datasets with the same poisoned ratios (p = 0.3%) or same number (r = 18) of backdoor samples.

### C.2   ATTACK SUCCESS RATE WITH SMALLER TRIGGER SIZE

Typically, to ensure stealth, attackers aim to minimize the amount of poisoning while maintaining a high success rate for the attack. Our preliminary findings indicate that in datasets with extreme imbalance, a poisoning rate of $0.3\%$ in step imbalance can achieve an ASR of more than $95\%$, as shown in Fig. 1.

As demonstrated in Thm. 5.1, a sample attacked by a backdoor is guaranteed to be detected with trigger size $\|\delta\|_2 \geq R = \sigma \left( \Phi^{-1}(p(x) - \zeta(x, \delta)) - \Phi^{-1}(\overline{p_t}) \right)$. In this section, we examine scenarios where the trigger size is smaller than $R_{\sigma, t}$, while keeping the poisoning rate constant. As shown in Tab. 4, all evaluated attacks struggle to maintain high ASR under such stealthy constraints, confirming the general difficulty of successful backdoor injection when both poisoning budget and perturbation size are limited. In practice, defensive measures are not required when the ASR drops below $50\%$.

| Attack | $\rho = 1$ | | $\rho = 10$ | | $\rho = 200$ | |
|---|---|---|---|---|---|---|
| | AUC | ASR | AUC | ASR | AUC | ASR |
| Badnets | 100.0 | 13.8 | 100.0 | 14.5 | 98.9 | 21.8 |
| Blend | 100.0 | 16.3 | 100.0 | 18.8 | 99.0 | 22.9 |
| Trojan | 100.0 | 29.4 | 100.0 | 30.6 | 99.0 | 35.6 |
| SIG | 100.0 | 29.0 | 100.0 | 33.7 | 99.0 | 38.2 |
| ISSBA | 99.8 | 30.2 | 99.8 | 47.6 | 98.8 | 53.3 |
| WaNet | 100.0 | 35.8 | 99.7 | 46.0 | 98.9 | 55.1 |
| Sleeper Agent | 99.9 | 28.6 | 99.5 | 36.9 | 99.0 | 48.6 |
| Adaptive patch | 99.8 | 20.1 | 99.8 | 26.9 | 98.5 | 33.0 |
| Adaptive Blend | 99.9 | 22.7 | 99.0 | 26.1 | 98.9 | 30.4 |

Table 4: The ASR and AUC under $\|\delta\|_2 < 0.8$ on the balanced MNIST dataset ($\rho = 1$) and two types of imbalanced MNIST datasets ($\mu = 0.9$, $\rho = 10, 200$) with same poisoned ratio (p=0.3%).

## C.3 RELATIVE FREQUENCY DISTRIBUTIONS OF RPP

In Sec. 4.1, we present the relative frequency distributions of $\tilde{\Delta}P$ for clean and malicious samples under datasets with imbalance ratios $\rho = 2$ and $\rho = 100$. In Fig. 9, the relative frequency distribution of $\tilde{\Delta}P$ for clean and malicious samples in balanced datasets ($\rho = 1$) as well as those with imbalance ratios $\rho = 10$ and $\rho = 200$ are depicted. It is observed that images subjected to backdoor attacks typically exhibit a lower $\tilde{\Delta}P$ compared to clean images. Although there is some overlap in the $\tilde{\Delta}P$ between poisoned data and clean samples when $\rho = 200$, the RPP method still achieves TPR of 90.0% with FPR below 30% (Fig. 4). Moreover, datasets with a step imbalance of $\rho = 200$ are also rare in real-world scenarios.

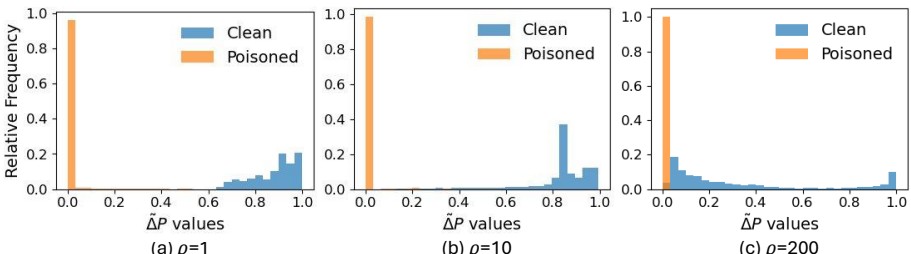

Figure 9: Relative frequency distribution of $\tilde{\Delta}P$ for clean and poisoned samples on SVHN dataset with imbalance ratios: (a) $\rho = 1$, (b) $\mu = 0.9$, $\rho = 10$ and (c) $\mu = 0.9$, $\rho = 200$.

## C.4 EMPIRICAL VALIDATION FOR ASSUMPTION (A.1)

To empirical validate the Assumption (A.1): $p_{y_t}(x + \delta + \varepsilon \mid w) < p_{y_t}(x + \delta \mid w)$, we evaluated the percentage of backdoored test samples satisfying this inequality on CIFAR-10 under varying imbalance ratios, with Gaussian noise ($\sigma$=1.0). Tab. 5 showed the assumption is consistent in all settings. Further, we tested backdoor-triggered samples with and without Gaussian noise across various imbalance settings. Tab. 6 clearly demonstrate that as the noise level $\sigma$ increases, the ASR decreases across all imbalance ratios. This confirms that noise-perturbed poisoned samples are indeed less likely to be classified into the target class, providing strong empirical support for our theoretical assumption.

## C.5 PERFORMANCE OF DIFFERENT NOISE SAMPLES

In Fig. 10, we evaluate the performance of RPP across different numbers of Gaussian noise samples $J$, which are used to estimate classification probability variations. The results demonstrate that RPP achieves over 95% detection performance on datasets with imbalance ratios $\rho = 1, 2, 10$, and 100. Even under a high imbalance ratio of $\rho = 200$, RPP maintains a TPR exceeding 85% while keeping the FPR below 40%. Notably, the results indicate that a small number of noise samples (as few as

| Attack | $\rho = 1$ | $\rho = 2$ | $\rho = 10$ | $\rho = 100$ | $\rho = 200$ |
|---|---|---|---|---|---|
| Badnets | 100.0 | 100.0 | 100.0 | 100.0 | 99.8 |
| Blend | 100.0 | 100.0 | 100.0 | 100.0 | 99.6 |
| Trojan | 100.0 | 100.0 | 100.0 | 100.0 | 99.7 |
| Narcissus | 100.0 | 100.0 | 100.0 | 100.0 | 100.0 |

Table 5: Percentage of poisoned samples satisfying Assumption 1 under different imbalance ratios $\rho$ with $\sigma = 1.0$ on CIFAR-10, evaluated on four backdoor attacks: Badnets, Blend, Trojan, and Narcissus.

| | $\sigma = 0$ | $\sigma = 0.5$ | $\sigma = 1.0$ | $\sigma = 1.5$ |
|---|---|---|---|---|
| $\rho = 1$ | 100.0 | 84.6 | 77.4 | 68.3 |
| $\rho = 2$ | 99.9 | 83.7 | 77.7 | 70.3 |
| $\rho = 10$ | 99.7 | 94.8 | 89.3 | 83.9 |
| $\rho = 100$ | 100.0 | 98.6 | 94.4 | 90.9 |
| $\rho = 200$ | 100.0 | 99.1 | 97.5 | 93.1 |

Table 6: The ASR of Badnets backdoor attack was evaluated on the balanced CIFAR10 training dataset ($\rho = 1$) and imbalanced CIFAR10 training datasets ($\rho = 2, 10, 100, 200$) with poisoning ratio (p = 0.3%) under different standard deviations of isotropic Gaussian noise ($\sigma = 0, 0.5, 1.0, 1.5$).

$J = 3$) is sufficient to achieve strong detection performance. To balance detection effectiveness and computational efficiency, we set $J = 3$ in all subsequent experiments.

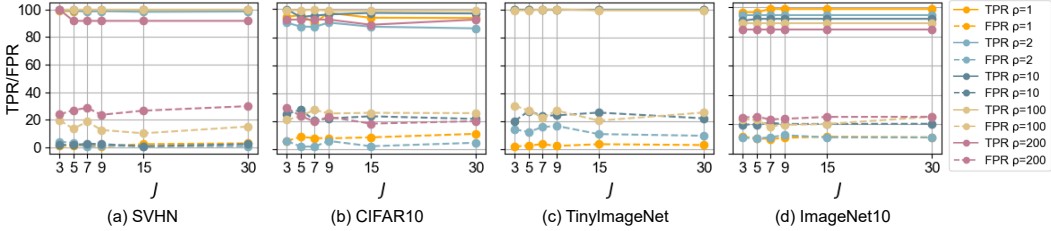

Figure 10: Performance of RPP against Badnets backdoor attack with perturbation magnitude $\|\delta\|_2 \geq 0.8$ across balanced dataset ($\rho = 1$) and varying imbalance ratios ($\mu = 0.9$, $\rho = 2, 10, 100, 200$) under different J with $n = 100$, $\alpha = 0.05$ on SVHN, CIFAR-10, TinyImageNet and ImageNet10 datasets.

### C.6 PERFORMANCE ON DIFFERENT MODEL ARCHITECTURES

In Tab. 7, we present the performance of RPP across different model architectures such as VGG16, DenseNet161, and EfficientNetB0 under varying imbalance ratios ($\rho$). RPP achieves consistently high TPR across all settings, where TPRs often reach or approach 100%. Notably, RPP maintains a strong detection capability even under severe imbalance ($\rho = 100$ or $\rho = 200$), demonstrating its robustness across datasets and architectures. TinyImageNet exhibits a similar trend but tends to yield lower FPRs under balanced conditions and slightly higher TPRs under extreme imbalance compared to CIFAR10.

### C.7 IMPACT OF CALIBRATION SET SIZE

The selection of the calibration set size ($n$) significantly impacts the performance of conformal prediction. Intuitively, a larger $n$ may seem preferable as it generally results in more stable procedures, a notion supported by (Vovk, 2012). Generally speaking, choosing a calibration set of size $n = 1000$ is sufficient for most purposes (Angelopoulos et al., 2023). However, in practical scenarios, it is challenging to obtain an additional and big clean calibration set ($n = 1000$) that is identically distributed (i.i.d.) with the original data set. In Fig. 11, we present the TPR and FPR of RPP against the Badnets attack across three datasets, evaluated with different calibration set sizes ($n$=100, 200, 400, 600, 800 and 1000) and $\alpha = 0.05$. Our findings demonstrate that RPP's computational and

| | CIFAR10 | | | | | | TinyImageNet | | | | | |
| | VGG16 | | DenseNet161 | | EfficientNetB0 | | VGG16 | | DenseNet161 | | EfficientNetB0 | |
| | TPR | FPR | TPR | FPR | TPR | FPR | TPR | FPR | TPR | FPR | TPR | FPR |
|---|---|---|---|---|---|---|---|---|---|---|---|---|
| $\rho = 1$ | 99.9 | 10.1 | 94.4 | 18.3 | 90.3 | 13.7 | 100.0 | 4.2 | 100.0 | 9.6 | 92.2 | 7.8 |
| $\rho = 2$ | 99.9 | 24.3 | 92.6 | 24.6 | 89.8 | 12.8 | 99.9 | 9.9 | 86.9 | 16.7 | 96.6 | 13.3 |
| $\rho = 10$ | 93.5 | 29.8 | 87.5 | 22.0 | 86.6 | 25.5 | 100.0 | 29.3 | 91.8 | 23.7 | 100.0 | 31.0 |
| $\rho = 100$ | 87.1 | 22.8 | 86.7 | 29.7 | 81.8 | 23.3 | 100.0 | 36.6 | 82.2 | 27.8 | 88.7 | 28.2 |
| $\rho = 200$ | 90.0 | 26.9 | 88.9 | 30.6 | 77.8 | 23.8 | – | – | – | – | – | – |

Table 7: Performance of RPP against Badnets backdoor attack with perturbation magnitude $\|\delta\|_2 \geq 0.8$ across balanced dataset ($\rho = 1$) and varying imbalance ratios ($\mu = 0.9$, $\rho = 2, 10, 100, 200$) on CIFAR10 and TinyImageNet datasets under VGG16, DenseNet161, and EfficientNetB0 architectures with $n = 100$, $\alpha = 0.05$, and $\sigma = 1.0$ for CIFAR10 and TinyImageNet.

data efficiency can be enhanced by reducing the size of the calibration set, without significantly compromising its detection or certification performance. Specifically, when the calibration set size is decreased from 1000 to 100, the TPR remains nearly unchanged across datasets with varying degrees of imbalance. This robustness of RPP to the calibration set size further validates the clear separation between benign and backdoored samples, underscoring its effectiveness in practical scenarios.

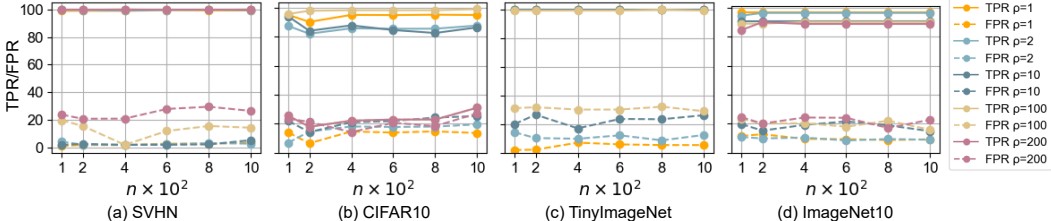

Figure 11: Performance of RPP against Badnets backdoor attacks with perturbation magnitude $\|\delta\|_2 \geq 0.8$, measured by TPR and FPR across balanced dataset ($\rho = 1$), and varying imbalance ratios ($\mu = 0.9$, $\rho = 2, 10, 100, 200$) for a range of $n$ with $\alpha = 0.05$ on SVHN, CIFAR-10, TinyImageNet and ImageNet10 datasets.

## C.8 PERFORMANCE ON MNIST

We use a standard 2-layer convolutional neural network (App. B.3) for MNIST and report TPR/FPR under a balanced setting ($\rho = 1$) and multiple imbalance ratios ($\rho = 2, 10, 100, 200$). Fig. 12 shows as $\alpha$ increases from 0.05 to 0.10, TPR remains near ceiling across all $\rho$, while FPR rises slightly yet stays modest. Sweeping $\sigma$ from 0.05 to 0.5 yields TPR close to 100% for most $\rho$, with only mild degradation at the largest $\sigma$ under severe imbalance; FPR typically remains below 35%. Increasing $n$ from 100 to 1000 keeps TPR essentially saturated and generally reduces FPR, indicating that larger calibration sets stabilize false-positive control. Finally, varying the number of Gaussian noise samples $J$ maintains TPR around 100% across $\rho$.

## C.9 PERFORMANCE ON iNATURALIST

To further evaluate the generalization of RPP in realistic long-tailed settings, we conduct experiments on the naturally imbalanced iNaturalist-2018 dataset (Van Horn et al., 2018). Ten classes are randomly selected from the training data while preserving the dataset's inherent long-tailed distribution, forming an iNaturalist-10 subset. We employ a ViT-B/16 model as the classifier and evaluate RPP against multiple backdoor attack types with $n = 100$, $\alpha = 0.05$, and $\sigma = 1.0$. As summarized in Table 8,

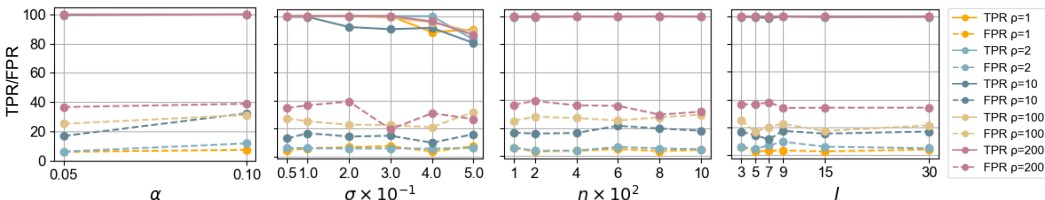

Figure 12: Performance of the RPP detection against Badnets backdoor attacks with perturbation magnitude $\|\delta\|_2 \geq 0.8$, measured by TPR and FPR across balanced dataset ($\rho = 1$), and varying imbalance ratios ($\mu = 0.9$, $\rho = 2, 10, 100$) on MNIST dataset. (a) For different $\alpha$ values with $\sigma = 0.1$ and $n = 100$. (b) For a range of $\sigma$ values with $\alpha = 0.05$ and $n = 100$. (c) For various calibration size $n$ with $\alpha = 0.05$ and $\sigma = 0.1$.

|     | Badnets | Blend | Trojan | SIG  | ISSBA | WaNet |
|-----|---------|-------|--------|------|-------|-------|
| TPR | 81.1    | 82.2  | 74.8   | 77.0 | 78.5  | 87.1  |
| FPR | 18.7    | 17.2  | 15.6   | 10.9 | 24.3  | 12.8  |

Table 8: Detection performance (TPR/FPR, %) of RPP against different backdoor attack types with $n = 100$, $\alpha = 0.05$, and $\sigma = 1.0$ on the iNaturalist dataset.

RPP maintains comparable detection performance on this naturally imbalanced dataset, highlighting its robustness.

## C.10 Performance under Imbalanced Calibration Set

In Tab. 9, we present the detection performance of RPP with $\alpha = 0.05$ when the validation set consists of imbalanced clean data that are i.i.d. with the training set. For the imbalanced calibration set, we include 100 samples from the majority class and 1 sample from each minority class, resulting in $n = 109$ for MNIST, SVHN, and CIFAR-10, and $n = 149$ for TinyImageNet. The results suggest that the detection efficacy of RPP is unaffected by whether the calibration set is balanced or imbalanced.

|              | $\rho = 1$ | | $\rho = 2$ | | $\rho = 10$ | | $\rho = 100$ | | $\rho = 200$ | |
|--------------|-------|------|-------|------|-------|------|-------|------|-------|------|
|              | TPR   | FPR  | TPR   | FPR  | TPR   | FPR  | TPR   | FPR  | TPR   | FPR  |
| MNIST        | 100.0 | 5.7  | 100.0 | 6.9  | 82.4  | 0.5  | 100.0 | 1.9  | 100.0 | 6.7  |
| SVHN         | 99.7  | 11.6 | 98.8  | 3.6  | 100.0 | 5.9  | 98.9  | 2.9  | 89.7  | 4.6  |
| CIFAR10      | 98.0  | 5.8  | 93.1  | 3.0  | 97.4  | 7.0  | 86.1  | 3.8  | 87.0  | 12.4 |
| TinyImageNet | 100.0 | 1.7  | 100.0 | 2.9  | 98.0  | 2.6  | 85.7  | 4.7  | –     | –    |

Table 9: Performance of the RPP detection against Badnets backdoor attacks with perturbation magnitude $\|\delta\|_2 \geq 0.8$, measured by TPR and FPR across balanced dataset ($\rho = 1$) and varying imbalance ratios ($\mu = 0.9$, $\rho = 2, 10, 100, 200$) with $\alpha = 0.05$ on MNIST, SVHN, CIFAR-10, and TinyImageNet datasets.

## C.11 Resistance to Potential Adaptive Attacks

In realistic attack scenarios, attackers typically minimize the poisoning rate to preserve stealth while still achieving a high ASR. This assumption is well grounded in the backdoor literature; for example, (Souri et al., 2022; Zeng et al., 2023) adopt triggers with a norm $\ell_\infty$ bounded by 16/255 to maintain a high ASR under a poisoning budget of 0.5% or 1%. Following this common assumption, we adopt low poisoning rates in all experiments to reflect realistic threat settings. Fig. 1 demonstrates that with a trigger satisfying $\|\delta\|_2 \geq 0.8$, a 0.3% poisoning rate under step imbalance can yield an ASR exceeding 95%. In contrast, using smaller triggers under the same poisoning rate drops the ASR to 20% (App. C.2), highlighting the necessity of slightly larger perturbation norms for maintaining attack effectiveness in low-poisoning regimes.

|     | $\rho{=}1$ | $\rho{=}2$ | $\rho{=}10$ | $\rho{=}100$ | $\rho{=}200$ |
|-----|------|------|------|------|------|
| TPR | 99.7 | 92.0 | 76.4 | 71.0 | 46.5 |
| FPR | 15.9 | 12.4 | 22.6 | 22.8 | 23.0 |

Table 10: Performance of RPP against Badnets across class-imbalance ratios $\rho$ under $\|\delta\|_2 < 0.8$ on CIFAR-10.

|     | $\rho{=}1$ | $\rho{=}2$ | $\rho{=}10$ | $\rho{=}100$ | $\rho{=}200$ |
|-----|------|------|------|------|------|
| TPR | 90.0 | 81.5 | 79.0 | 61.8 | 59.7 |
| FPR | 6.8  | 13.1 | 12.5 | 16.9 | 26.4 |

Table 11: Performance of RPP against Badnets adaptive attack across class-imbalance ratios $\rho$ on CIFAR-10.

We probe robustness und er a worst-case adaptive setting in which the adversary has full knowledge of our defense and deliberately chooses smaller triggers with $\|\delta\|_2 < 0.8$. To preserve high ASR in this regime, the attacker must increase the poisoning rate (we set $p = 10\%$), achieving ASR $> 90\%$. Even under these adversarial conditions, RPP often empirically detects poisoned samples (Tab. 10). It is worthy mentioning that, when the trigger is extremely small, it is significantly more difficult to inject backdoor with a reasonable attack buddget, thus not practical in many threat scenarios.

We also consider a different form of adaptive attack, termed the *hybrid-label adaptive attack*, where the adversary, aware of our detection mechanism, poisons 80% of samples via standard label flipping and 20% with triggers combined with Gaussian noise while keeping their original labels. This hybrid design introduces prediction instability in poisoned samples and aims to evade detection by inducing ambiguity in the predicted probabilities. Despite this, RPP remains robust, as shown in Tab. 11. The conformal prediction framework is calibrated exclusively on clean data, it adaptively rejects outliers that display irregular confidence fluctuations, thereby maintaining strong detection performance even under this hybrid-label adaptive scenario.

### C.12   ALGORITHM FOR RPP DETECTION AND CERTIFICATION

In this section, we present Algorithm 1, which details the procedure for identifying backdoored samples using the RPP detection method. The algorithm accepts an input $x$ from $\mathcal{D}(x + \delta)$ to be inspected and a clean calibration dataset, $\mathcal{V}$. By computing the empirical $\tilde{\Delta}P(x \mid w, \sigma)$, over $J$ noise-injected samples. Then compared $\tilde{\Delta}P(x \mid w, \sigma)$ against a threshold derived from the calibration set $\mathcal{S}$, which consists of RPP values calculated for each instance in $\mathcal{V}$. The threshold is determined by finding the quantile specified by the significance level $\alpha$ and the calibration set size of $n$. The sample is classified as poisoned if its RPP is less than or equal to this quantile threshold, otherwise, it is deemed clean.

---

**Algorithm 1** Identification of Backdoored Samples

---

1: **Input:** $x$: data from $\mathcal{D}(x + \delta)$ to be inspected, $\delta$: backdoor trigger, $\mathcal{V}$: clean calibration data set.
2: **Parameters:** $n$: size of $\mathcal{V}$, $\sigma$: noise standard deviation, $J$: number of noise injections, $\alpha$: significance level.
3: **Step 1:** Pretrain a classifier $f(\cdot \mid w)$ using $\mathcal{D}(x + \delta)$; Compute the empirical RPP $\tilde{\Delta}P(x \mid w, \sigma)$ over $J$ samples as defined in eq. (2).
4: **Step 2:** Form the calibration set $\mathcal{S} = \{s_1, \ldots, s_n\}$, where $s_i = \tilde{\Delta}P(x_i \mid w, \sigma)$ for each $x_i \in \mathcal{V}$.
5: **Step 3:** Find the $\frac{\lceil \alpha(n+1) \rceil}{n}$ quantile as

$$\hat{q} = \inf \left\{ q : \frac{|\{i : s_i \leq q\}|}{n} \geq \frac{\lceil \alpha(n+1) \rceil}{n} \right\}$$

6: **Output:** Classify $x$ as poisoned if $\tilde{\Delta}P(x \mid w, \sigma) \leq \hat{q}$; otherwise, classify $x$ as clean.

---

In Algorithm 2, we outline a method for certifying the detection of poisoned samples using the RPP approach. Estimating the probability $p(x)$ that the modified data point $x + \delta$ is classified as a target class $y_t$ under the pre-trained classifier $f(\cdot \mid w)$. We estimate $\zeta(x, \delta)$ using the $\frac{\lceil \alpha(n+1) \rceil}{n}$-quantile of the calibration set $\mathcal{S}$ and let $y_t$ denote the class that appears most frequently among $J$ noise-injected copies of $x$, where $\epsilon_1, \ldots, \epsilon_J \sim \mathcal{N}(0, \sigma^2 I)$. Specifically, for each $x + \epsilon_j$, we process the sample through the classifier $f(\cdot \mid w)$, tally the frequency of each class across all $J$ predictions, and identify $y_t$ as the class with the highest occurrence. The count of occurrences for $y_t$ within these $J$ samples, denoted as $n_t = \text{counts}[y_t]$, is used to determine the upper confidence bound for this probability, taking into account the noise injections and the significance level $\alpha$. If the calculated bounds on $\delta$ allow for guaranteed detection of the poisoned samples, providing a clear criterion for certifying the presence of backdoors.

---

**Algorithm 2** RPP Certification

---

1: **Input:** $x$: data point to be inspected, $\mathcal{V}$: clean calibration data set, $f(\cdot \mid w)$: pre-trained classifier, $\delta$: backdoor trigger, $\Phi$: a standard Gaussian CDF.
2: **Parameters:** $n$: size of $\mathcal{V}$, $\sigma$: noise standard deviation, $J$: number of noise injections, $\alpha$: significance level.
3: $p(x) \leftarrow \mathbb{P}(f(x + \delta \mid w) = y_t)$
4: Get $\zeta(x, \delta) \leftarrow \frac{\lceil \alpha(n+1) \rceil}{n}$ quantile from the calibration set $\mathcal{S}$
5: $y_t \leftarrow$ top class in $f(x + \varepsilon_j \mid w)$
6: $n_t \leftarrow \text{counts}[y_t]$
7: $\overline{p_t} \leftarrow \text{UPPERCONFBOUND}(n_t, J, 1 - \alpha)$
8: **Output:** If $\sigma\left(\Phi^{-1}(p(x) - \zeta(x, \delta)) - \Phi^{-1}(\overline{p_t})\right) \leq \|\delta\|_2 \leq \sigma\left(\Phi^{-1}(p(x)) - \Phi^{-1}(\overline{p_t})\right)$, the poisoned samples are guaranteed to be detected; otherwise, the poisoned samples may be detected but without guarantee.

---

C.13    BASELINES FOR BACKDOOR SAMPLE DETECTION

We compare our method against 12 empirical backdoor defenses on both balanced ($\rho = 1$) and imbalanced CIFAR-10 datasets ($\rho = 10, 200$), as summarized in Tab. 12. Detection results for RPP are reported under $\alpha = 0.05$, $n = 100$, and $\sigma = 1.0$, across 10 types of backdoor attacks. In all cases, we ensure ASR exceeds 90%. RPP consistently achieves comparable or superior TPRs relative to SOTA defenses across all trigger types. We also briefly review these baseline methods and analyze why they fall short in challenging imbalanced settings.

SS(Tran et al., 2018) uses singular value decomposition to detect anomalies by selecting a subset of data comprising 1.5 times the number of samples with the highest anomaly scores. However, the anomaly detection strategies struggle when the number of poisoned samples is extremely low and the dataset is imbalanced. AC (Chen et al., 2018) captures activation data from the terminal hidden layer of a trained neural network and applies clustering algorithms to pinpoint and expunge anomalous instances from the training dataset. However, the method's efficacy is limited due to the uniform clustering effect and the substantial ratio disparity between poisoned and clean samples, which hinders its performance. ABL(Li et al., 2021a) was initially developed as a robust training defense and subsequently repurposed as a detection method based on output losses. This approach exhibits limited efficacy due to the significantly lower isolation rate of poisoned samples compared to the actual poisoning rate. With only a minimal number of poisoned samples isolated, often as few as two or three, the effectiveness of this method is substantially compromised. CT (Qi et al., 2023b) utilized confusion training for detecting backdoored samples. When the dataset exhibits extreme imbalance and the target label corresponds to the majority class, clean samples are predominantly predicted as belonging to this majority class, resulting in low training losses for these clean samples. ASSET (Pan et al., 2023): In the imbalanced datasets and low poison ratio settings, when training with mini-batches, the number of poisoned samples in each mini-batch is very small, or even nonexistent. In the inner offset loop, different mini-batches may contain different amounts of poisons and even no poisoned samples, so it is hard to determine the number of most suspicious samples within each mini-batch; Additionally, when using the gradient ascent algorithm, the optimization process tends to overlook the characteristics of these few samples. This means that the model may not adequately learn the features of these poisoned samples during the learning process. SCALE-UP (Guo et al., 2023): When

the dataset is imbalanced, although the model still exhibits some consistency in classifying backdoor samples, the bias towards the majority class also results in consistent classification predictions for clean samples, which leads to a high rate of misdiagnosis. MSPC (Pal et al., 2024) employs a bi-level optimization technique designed to minimize the cross-entropy loss for clean samples while simultaneously maximizing it for backdoor samples, facilitating the identification of backdoor samples. However, when the dataset is highly imbalanced and the poisoning rate is low, the use of upper-level optimization to increase the loss for backdoor samples yields suboptimal results. STRIP (Gao et al., 2019) relies on entropy-based thresholds, which become unreliable when minority-class benign samples naturally exhibit low entropy, leading to higher false positives. TED (Mo et al., 2024) assumes stable topological evolution for benign samples, but class imbalance distorts layer-wise activation patterns, making poisoned and minority-class samples harder to distinguish. BBCaL (Hu et al., 2024) uses random perturbations to probe behavior changes, but its fixed calibration thresholds are not robust across imbalanced distributions, reducing detection sensitivity. IBD-PSC (Hou et al., 2024) detects backdoor presence based on maximum margin statistics, but this method becomes unstable under severe imbalance due to a similar overfitting phenomenon and biased logit distributions across classes. RE (Xiang et al., 2021) introduces an optimization-based defense that detects backdoors, identifies the target class, and removes poisoned samples by estimating minimal source-to-target perturbations. However, its performance degrades under data imbalance, since minority classes yield unstable perturbation estimates and biased anomaly statistics, making the inferred backdoor signal less distinguishable.

| | | Badnets | | Blend | | Trojan | | SIG | | ISSBA | | WaNet | | Sleeper Agent | | AdaPatch | | AdaBlend | | Narci. | |
|---|---|---|---|---|---|---|---|---|---|---|---|---|---|---|---|---|---|---|---|---|---|
| | Defenses | TPR | FPR | TPR | FPR | TPR | FPR | TPR | FPR | TPR | FPR | TPR | FPR | TPR | FPR | TPR | FPR | TPR | FPR | TPR | FPR |
| $\rho=1$ | SS | 93.1 | 5.6 | 91.8 | 4.9 | 60.2 | 10.7 | 61.0 | 20.1 | 45.5 | 4.2 | 59.6 | 8.8 | 61.3 | 5.1 | 85.9 | 6.7 | 83.6 | 8.2 | 12.1 | 8.5 |
| | AC | 97.8 | 2.8 | 96.4 | 5.7 | 73.6 | 16.0 | 78.9 | 10.4 | 51.2 | 7.0 | 23.1 | 10.6 | 45.6 | 17.9 | 91.7 | 6.6 | 88.0 | 9.3 | 9.3 | 26.7 |
| | RE | 92.1 | 10.3 | 90.8 | 8.9 | 88.2 | 12.4 | 64.2 | 9.5 | 80.1 | 11.9 | 78.5 | 15.0 | 66.4 | 9.0 | 82.3 | 10.4 | 80.5 | 7.5 | 14.3 | 5.8 |
| | ABL | 86.4 | 3.9 | 90.2 | 10.3 | 85.0 | 4.8 | 85.1 | 8.3 | 52.8 | 3.5 | 78.6 | 8.2 | 20.3 | 1.7 | 55.8 | 3.6 | 50.2 | 3.9 | 1.8 | 2.8 |
| | CT | 93.9 | 8.7 | 95.2 | 6.3 | 88.9 | 5.0 | 92.0 | 19.6 | 95.2 | 11.0 | 92.6 | 9.5 | 82.1 | 11.2 | 84.0 | 4.3 | 86.7 | 6.5 | 22.1 | 17.5 |
| | ASSET | 89.9 | 17.8 | 83.1 | 16.0 | 90.4 | 18.3 | 86.7 | 20.0 | 91.2 | 8.8 | 93.0 | 11.7 | 52.9 | 15.4 | 55.7 | 21.3 | 80.3 | 10.6 | 92.1 | 4.3 |
| | SCALE-UP | 100.0 | 19.3 | 77.1 | 20.1 | 67.4 | 18.8 | 98.8 | 32.3 | 92.5 | 21.2 | 84.2 | 19.9 | 45.1 | 8.5 | 80.7 | 30.0 | 79.9 | 22.4 | 67.3 | 19.5 |
| | MSPC | 92.2 | 22.7 | 94.8 | 12.4 | 66.7 | 16.0 | 91.4 | 22.3 | 89.5 | 15.5 | 94.9 | 8.4 | 79.1 | 21.2 | 80.3 | 29.6 | 84.4 | 18.5 | 82.4 | 23.6 |
| | BBCaL | 98.9 | 10.1 | 96.8 | 12.4 | 80.3 | 9.5 | 98.3 | 13.1 | 99.7 | 9.6 | 90.2 | 14.2 | 91.0 | 16.6 | 75.1 | 15.7 | 72.7 | 13.6 | 86.6 | 9.4 |
| | STRIP | 98.5 | 10.3 | 84.4 | 15.6 | 33.1 | 11.0 | 94.3 | 22.1 | 48.6 | 19.7 | 13.3 | 29.9 | 22.6 | 16.1 | 88.8 | 19.6 | 87.6 | 20.5 | 0.0 | 0.8 |
| | TED | 100.0 | 2.9 | 94.3 | 6.4 | 80.6 | 9.3 | 91.1 | 13.6 | 93.6 | 12.5 | 92.6 | 8.8 | 90.3 | 11.4 | 88.4 | 10.3 | 86.2 | 13.5 | 91.7 | 10.5 |
| | IBD-PSC | 99.4 | 4.5 | 97.9 | 5.1 | 93.0 | 8.6 | 87.2 | 10.4 | 98.6 | 8.7 | 95.6 | 7.3 | 69.0 | 9.5 | 92.0 | 10.1 | 86.7 | 9.6 | 93.1 | 24.7 |
| | Ours | 98.5 | 5.9 | 98.8 | 6.1 | 92.3 | 9.6 | 98.5 | 3.2 | 94.6 | 7.7 | 93.7 | 6.5 | 95.0 | 16.2 | 92.2 | 6.0 | 89.0 | 10.4 | 96.1 | 19.6 |
| $\rho=10$ | SS | 35.2 | 7.3 | 37.7 | 8.2 | 35.3 | 4.1 | 60.7 | 0.2 | 15.1 | 10.0 | 38.7 | 6.6 | 40.9 | 31.2 | 22.5 | 3.7 | 13.3 | 6.6 | 0.0 | 0.6 |
| | AC | 50.8 | 3.8 | 34.1 | 4.8 | 17.9 | 3.0 | 60.0 | 36.1 | 35.5 | 16.8 | 50.2 | 12.4 | 20.1 | 16.2 | 42.5 | 43.6 | 20.8 | 4.9 | 0.0 | 0.0 |
| | RE | 41.6 | 18.8 | 40.3 | 15.5 | 43.7 | 13.0 | 29.6 | 15.4 | 39.6 | 17.4 | 35.0 | 12.6 | 25.9 | 11.4 | 40.7 | 14.5 | 37.6 | 13.0 | 13.9 | 17.6 |
| | ABL | 0.0 | 0.2 | 0.0 | 0.0 | 0.0 | 0.0 | 0.0 | 0.0 | 9.2 | 2.7 | 47.5 | 7.3 | 0.0 | 2.6 | 0.6 | 0.2 | 0.0 | 0.3 | 0.0 | 0.8 |
| | CT | 68.9 | 0.9 | 60.4 | 2.9 | 60.4 | 8.9 | 59.8 | 0.1 | 75.9 | 8.9 | 80.6 | 13.9 | 26.1 | 0.0 | 0.0 | 0.0 | 32.5 | 22.4 | 0.0 | 8.9 |
| | ASSET | 60.0 | 6.9 | 51.2 | 18.4 | 39.3 | 41.5 | 10.6 | 19.7 | 78.6 | 18.8 | 69.5 | 14.1 | 71.4 | 15.0 | 38.3 | 31.1 | 40.6 | 31.3 | 51.7 | 13.9 |
| | SCALE-UP | 95.9 | 84.3 | 67.3 | 35.1 | 67.2 | 41.5 | 71.1 | 34.0 | 86.2 | 22.6 | 78.0 | 19.6 | 47.3 | 21.5 | 68.3 | 32.5 | 62.9 | 37.7 | 39.0 | 12.0 |
| | MSPC | 89.6 | 16.5 | 89.5 | 11.3 | 55.7 | 21.6 | 83.9 | 10.0 | 80.1 | 20.3 | 76.9 | 9.4 | 63.7 | 0.9 | 51.2 | 24.3 | 41.0 | 9.5 | 49.6 | 29.4 |
| | BBCaL | 83.2 | 17.9 | 88.7 | 22.0 | 67.3 | 19.8 | 79.7 | 14.5 | 78.6 | 21.3 | 72.2 | 20.7 | 69.4 | 25.3 | 60.1 | 22.4 | 59.9 | 25.6 | 66.4 | 15.0 |
| | STRIP | 63.4 | 19.7 | 50.0 | 26.8 | 18.3 | 17.7 | 60.6 | 28.5 | 28.4 | 21.6 | 6.5 | 28.9 | 26.8 | 16.3 | 64.8 | 26.8 | 44.3 | 28.8 | 0.0 | 0.5 |
| | TED | 90.4 | 12.2 | 82.5 | 14.3 | 54.3 | 17.6 | 72.0 | 16.9 | 79.9 | 21.0 | 76.5 | 16.4 | 70.1 | 19.4 | 66.8 | 21.1 | 61.2 | 20.3 | 76.9 | 18.6 |
| | IBD-PSC | 91.2 | 14.1 | 87.4 | 12.7 | 83.6 | 17.5 | 70.3 | 19.6 | 85.9 | 19.4 | 77.4 | 20.2 | 61.4 | 20.4 | 74.8 | 16.6 | 75.5 | 18.9 | 80.3 | 28.6 |
| | Ours | 96.7 | 3.2 | 85.2 | 10.4 | 96.4 | 9.1 | 80.0 | 4.2 | 88.1 | 13.6 | 87.5 | 11.7 | 78.7 | 19.5 | 82.4 | 3.3 | 82.6 | 12.5 | 83.7 | 20.1 |
| $\rho=200$ | SS | 0.1 | 0.0 | 4.9 | 0.7 | 14.7 | 4.9 | 0.0 | 5.0 | 4.4 | 8.1 | 7.1 | 6.0 | 0.0 | 0.1 | 7.5 | 1.4 | 9.5 | 3.6 | 0.0 | 0.3 |
| | AC | 0.0 | 0.0 | 0.0 | 0.5 | 0.0 | 1.0 | 0.0 | 9.4 | 0.6 | 2.4 | 0.0 | 5.1 | 0.2 | 0.0 | 0.0 | 0.0 | 0.0 | 0.0 | 0.0 | 0.0 |
| | RE | 16.4 | 7.0 | 12.8 | 7.4 | 8.6 | 7.8 | 10.6 | 0.0 | 6.9 | 12.2 | 10.5 | 11.5 | 0.0 | 10.3 | 0.0 | 0.7 | 2.1 | 1.9 | 0.0 | 6.2 |
| | ABL | 0.0 | 0.2 | 0.0 | 0.0 | 0.0 | 0.0 | 0.0 | 0.1 | 0.0 | 0.0 | 0.0 | 0.5 | 0.0 | 0.3 | 0.0 | 0.3 | 0.0 | 0.0 | 0.0 | 2.9 |
| | CT | 0.0 | 3.2 | 0.0 | 3.6 | 0.0 | 0.0 | 0.0 | 0.0 | 0.0 | 1.6 | 0.0 | 6.7 | 0.0 | 1.6 | 0.0 | 0.0 | 0.0 | 0.0 | 0.0 | 15.7 |
| | ASSET | 0.0 | 3.3 | 7.3 | 2.8 | 0.0 | 0.0 | 0.0 | 0.0 | 9.3 | 2.2 | 14.8 | 5.5 | 6.7 | 1.8 | 3.2 | 0.4 | 5.5 | 3.1 | 2.0 | 1.3 |
| | SCALE-UP | 67.3 | 54.1 | 51.4 | 48.3 | 0.0 | 0.0 | 0.0 | 0.0 | 33.8 | 19.9 | 48.6 | 30.1 | 0.2 | 42.1 | 6.3 | 0.6 | 29.7 | 30.1 | 10.1 | 21.8 |
| | MSPC | 10.6 | 2.8 | 20.2 | 6.1 | 15.3 | 3.1 | 19.2 | 2.9 | 18.8 | 5.6 | 32.6 | 17.3 | 10.6 | 5.8 | 8.2 | 1.7 | 9.4 | 3.2 | 21.7 | 32.8 |
| | BBCaL | 41.5 | 33.6 | 42.2 | 35.1 | 30.6 | 26.7 | 37.0 | 35.8 | 30.1 | 31.9 | 29.3 | 36.2 | 49.2 | 27.7 | 22.8 | 28.6 | 23.0 | 29.1 | 39.3 | 18.9 |
| | STRIP | 18.6 | 20.1 | 20.4 | 31.4 | 0.0 | 12.7 | 22.6 | 33.8 | 9.5 | 21.7 | 0.3 | 8.4 | 1.2 | 10.5 | 21.0 | 30.6 | 10.3 | 21.8 | 0.0 | 1.2 |
| | TED | 46.7 | 12.6 | 44.9 | 19.7 | 12.7 | 18.3 | 50.4 | 22.1 | 46.8 | 14.5 | 29.7 | 22.7 | 30.2 | 22.6 | 27.5 | 28.9 | 50.2 | 20.4 | | |
| | IBD-PSC | 51.7 | 17.8 | 50.0 | 24.7 | 55.8 | 26.3 | 50.0 | 24.7 | 41.9 | 23.8 | 40.5 | 22.0 | 30.4 | 24.1 | 32.7 | 22.8 | 34.0 | 25.5 | 44.8 | 30.0 |
| | Ours | 83.3 | 14.3 | 75.5 | 19.8 | 58.2 | 9.1 | 50.5 | 9.1 | 57.1 | 10.6 | 56.8 | 23.3 | 69.7 | 3.8 | 42.8 | 6.2 | 41.5 | 10.2 | 56.7 | 23.2 |

Table 12: Comparison with SoTA defenses on balanced ($\rho = 1$) and imbalanced CIFAR-10 datasets ($\mu = 0.9$, $\rho = 10, 200$).

## C.14 CERTIFIED DEFENSES UNDER IMBALANCED DATASET

RAB (Weber et al., 2023), the inaugural robust training methodology, certifies its robustness against backdoor attacks via randomized smoothing. DPA (Levine & Feizi, 2020) offers a certifiable defense against general and label-flipping poisoning attacks by partitioning the training set into k segments. These methodologies provide either probabilistic or deterministic assurances that a model will produce the intended outputs under specific adversarial conditions or when subjected to a certain number of label flips. As shown in Tab. 13, the efficacy of these certified defenses tends to decline as the dataset becomes more imbalanced. For RAB, we configure $\|\delta\|_2 = 0.1$ with a smoothing parameter $\sigma = 1.0$ and a 2% poisoning ratio against Badnets attacks. For DPA, we established 50 partitions with 9 sample labels flipped.

|  | $\rho = 1$ | $\rho = 2$ | $\rho = 10$ | $\rho = 100$ | $\rho = 200$ |
|---|---|---|---|---|---|
| RAB | 41.5 | 40.2 | 38.8 | 21.0 | 15.7 |
| DPA | 56.2 | 46.1 | 26.6 | 22.7 | 14.8 |

Table 13: Certified accuracy (%) of RAB and DPA against Badnets attack on the balanced CIFAR-10 training dataset ($\rho = 1$) and imbalanced CIFAR-10 training datasets ($\mu = 0.9$, $\rho = 2, 10, 100, 200$).

### C.15 IMPACT OF TARGET LABEL CLASS

We evaluated the performance of our method when the backdoor target label was assigned to either a minority class (class 9) or an intermediate-frequency class (class 4). In all experiments, we ensured that the ASR remained above 90% to maintain a consistent attack strength. As shown in Tab. 14, the proposed RPP method consistently achieved high TPR across various imbalance ratios $\rho$, while maintaining low FPR.

|  | Minority class (9) | | Intermediate class (4) | |
|---|---|---|---|---|
|  | TPR | FPR | TPR | FPR |
| $\rho = 2$ | 99.4 | 7.3 | 93.9 | 3.1 |
| $\rho = 10$ | 93.4 | 10.1 | 96.2 | 3.3 |
| $\rho = 100$ | 99.9 | 2.9 | 99.3 | 1.5 |
| $\rho = 200$ | 90.0 | 6.0 | 79.6 | 4.6 |

Table 14: Performance of RPP against the BadNets backdoor attack on imbalanced CIFAR-10 training datasets ($\rho = 2, 10, 100, 200$) with a $0.3\%$ poisoning ratio, where the target class is either a minority class (class 9) or an intermediate-frequency class (class 4).

### C.16 PERFORMANCE UNDER LOGITS-ADJUSTED TRAINING

To address the performance degradation caused by class imbalance, we adopted logits adjustment (LA) (Menon et al., 2020) as a long-tailed training strategy. This technique explicitly calibrates the logits of each class based on its frequency, thereby mitigating the dominance of majority classes and enhancing the learning of minority-class features. While it slightly reduces ASR under severe imbalance, our RPP method still achieves high TPR (83–99%) across all settings, as shown in Tab. 15. The result demonstrates the robustness of our method under long-tailed training, and its ability to remain effective even when the backdoor attack becomes slightly weakened by class rebalancing techniques.

|  | ASR (Without LA) | ASR (With LA) | TPR | FPR |
|---|---|---|---|---|
| $\rho = 2$ | 100.0 | 99.9 | 99.2 | 22.2 |
| $\rho = 10$ | 99.8 | 99.4 | 96.2 | 20.3 |
| $\rho = 100$ | 100.0 | 97.4 | 97.8 | 26.1 |
| $\rho = 200$ | 99.7 | 94.3 | 83.0 | 10.4 |

Table 15: Performance of RPP against BadNets attack under logits adjustment on imbalanced CIFAR-10 datasets ($\rho = 2, 10, 100, 200$) with a poisoning ratio of 0.3.

### C.17 PERFORMANCE AGAINST DIFFERENT TRIGGER SIZES

In Tab. 16, we present the ASR and AUC metrics for various trigger sizes, and we evaluate the efficacy of RPP when the ASR exceeds 50%. Our findings indicate that for $\|\delta\|_2 = 3$, in imbalanced datasets with $\rho = 100, 200$, the ASR surpasses 50%, and our RPP achieves a TPR of 80% with an FPR of less than 30%. When $\|\delta\|_2 = 6$, for both the balanced dataset ($\rho = 1$) and imbalanced datasets ($\rho = 2, 10$), the ASR is above 50%, and RPP attains TPRs of 70.6%, 89.9%, and 87.4%, respectively, with an FPR below 15%. These results align well with our certification.

| | ‖δ‖₂ = 0.8 | | ‖δ‖₂ = 2 | | ‖δ‖₂ = 3 | | | | ‖δ‖₂ = 4 | | | | ‖δ‖₂ = 6 | | | | ‖δ‖₂ = 8 | | | |
|---|---|---|---|---|---|---|---|---|---|---|---|---|---|---|---|---|---|---|---|---|
| | No Defense | | No Defense | | No Defense | | RPP | | No Defense | | RPP | | No Defense | | RPP | | No Defense | | RPP | |
| | ASR | AUC | ASR | AUC | ASR | AUC | TPR | FPR | ASR | AUC | TPR | FPR | ASR | AUC | TPR | FPR | ASR | AUC | TPR | FPR |
| $\rho = 1$ | 14.3 | 100.0 | 33.7 | 100.0 | 35.6 | 100.0 | – | – | 39.2 | 100.0 | – | – | 52.3 | 100.0 | 73.1 | 4.1 | 90.5 | 100.0 | 97.2 | 8.0 |
| $\rho = 2$ | 14.8 | 100.0 | 37.8 | 100.0 | 33.5 | 100.0 | – | – | 47.9 | 100.0 | – | – | 69.6 | 100.0 | 88.3 | 6.9 | 90.3 | 100.0 | 96.0 | 6.1 |
| $\rho = 10$ | 15.4 | 100.0 | 30.5 | 100.0 | 38.3 | 100.0 | – | – | 49.4 | 100.0 | – | – | 84.8 | 100.0 | 90.1 | 9.6 | 99.8 | 100.0 | 97.9 | 7.7 |
| $\rho = 100$ | 23.0 | 99.3 | 39.6 | 99.4 | 52.1 | 98.6 | 73.8 | 10.1 | 100.0 | 99.4 | 87.5 | 10.7 | 100.0 | 99.6 | 95.2 | 10.1 | 100.0 | 99.5 | 100.0 | 10.9 |
| $\rho = 200$ | 36.0 | 92.7 | 49.2 | 92.1 | 68.0 | 96.2 | 89.5 | 14.8 | 99.5 | 97.5 | 96.2 | 15.3 | 100.0 | 97.1 | 100.0 | 20.1 | 100.0 | 97.8 | 100.0 | 16.4 |

Table 16: Performance of RPP under various trigger sizes with a $0.3\%$ poisoning ratio on the balanced MNIST ($\rho = 1$) and imbalanced MNIST datasets ($\mu = 0.9$, $\rho = 2, 10, 100, 200$) for $n = 100$ and $\alpha = 0.05$.

### C.18 VALIDATION OF THE FPR UPPER BOUND GUARANTEE

We validate the probabilistic upper bound for the FPR in Thm. 5.4 by conducting experiments on the CIFAR-10 dataset. Specifically, we first set aside $1,000$ clean images from CIFAR-10 as a validation pool. From this pool, we randomly sample $n \in \{100, 200, 400, 600\}$ images as the calibration dataset and the remaining images are used to evaluate the FPR. We consider two significance level $\alpha = 0.05$ and $\alpha = 0.1$. For each $(n, \alpha)$ pair, we repeat the experiments 300 times and report the $0.95$-quantile FPR in Tab. 17. The results show that the FPR in the validation set is bounded by the theoretically derived upper bound.

| | $\alpha = 0.05$ | | | | $\alpha = 0.1$ | | | |
|---|---|---|---|---|---|---|---|---|
| | $n = 100$ | $n = 200$ | $n = 400$ | $n = 600$ | $n = 100$ | $n = 200$ | $n = 400$ | $n = 600$ |
| $\rho = 1$ | 4.0 | 5.3 | 5.1 | 4.4 | 10.7 | 9.9 | 9.1 | 8.8 |
| $\rho = 2$ | 4.9 | 4.1 | 4.6 | 4.0 | 9.3 | 9.9 | 10.1 | 9.0 |
| $\rho = 10$ | 5.5 | 4.9 | 4.6 | 4.2 | 10.7 | 9.8 | 9.5 | 9.1 |
| $\rho = 100$ | 5.5 | 4.7 | 4.8 | 5.0 | 9.9 | 9.2 | 10.2 | 10.0 |
| $\rho = 200$ | 5.1 | 4.9 | 4.8 | 4.7 | 10.4 | 10.0 | 9.8 | 10.1 |
| **Upper bound** | **6.0** | **5.5** | **5.3** | **5.2** | **11.0** | **10.5** | **10.3** | **10.2** |

Table 17: The FPR (%) under varying imbalance ratio ($\rho$), isotropic Gaussian noise standard deviation ($\alpha$), and calibration set size ($n$) on CIFAR10 dataset.

## D DIFFERENCES BETWEEN RPP AND CBD

We delineate the key differences between RPP and CBD (Xiang et al., 2024) from both theoretical and methodological perspectives.

**Theoretical Perspective.** Although both RPP and CBD leverage randomized smoothing and draw on the Neyman–Pearson lemma as formal justification (as inspired by (Cohen et al., 2019)), the objects being certified and the proof strategies differ substantially:

- Theorem 4.1 in CBD finds the lower bound of $s(w)$ (i.e. $\ell_\infty$ norm of Local Dominant Probability), while we bound below the RPP (expectation of the difference probability vectors). These certified quantities reflect fundamentally different detection criteria.
- CBD applies part (1) of Lemma 4 from (Cohen et al., 2019)) to derive an *upper bound* on the response to backdoor triggers. RPP, by contrast, uitilize part (2) of Lemma 4 from (Cohen et al., 2019)) to construct a *lower bound* for reliably detecting poisoned samples. These choices lead to distinct forms of certification, with differing robustness implications.

**Methodological Setting (Threat Model).** CBD is designed for **post-training model inspection**: it aims to determine whether a given *model* has been compromised by a backdoor, without requiring access to the training data. In contrast, RPP addresses a more proactive threat model—**pre-training poisoned sample identification**. It operates directly on the training data, identifying and filtering out backdoor instances *prior* to any model training, enabling safe downstream learning even under stealthy poisoning attacks.

## E    COMPUTATIONAL COMPLEXITY

On a single NVIDIA A100-SXM4-80GB device, the detection time per sample varies across different datasets as follows: 0.006 seconds for MNIST, 0.0154 seconds for SVHN, 0.0162 seconds for CIFAR10, and 0.0261 seconds for TinyImageNet. And, on an NVIDIA GeForce RTX 4090 device, the detection times per sample for the same datasets are observed to be 0.021, 0.0272, 0.0281, and 0.0404 seconds, respectively.

## F    BROADER IMPACT

RPP offers a robust poisoned samples detection against backdoor threats in machine learning pipelines. It effectively maintain high performance even under challenging adversarial conditions, reducing the risk of malicious exploits while ensuring safer deployment of AI technologies. Consequently, this work has the potential to set new benchmarks in content moderation and ethical AI practices, fostering more secure and socially responsible applications across diverse domains.

## G    LIMITATIONS & FUTURE WORK

One potential limitation of our study lies the use of synthetically constructed imbalanced datasets. Although our approach considers both step and long-tailed imbalance types across a range of imbalance ratios, it may not fully reflect the inherent complexity and distributional characteristics of real-world imbalanced datasets. Future research could address this limitation by evaluating the proposed method on naturally imbalanced datasets to better assess its practical applicability and robustness. In addition, our theoretical analysis relies on the assumption of i.i.d. Gaussian perturbations. Extending this analysis to encompass non-Gaussian or correlated perturbations could offer a more comprehensive understanding of robustness under realistic conditions and serves as a promising direction for future work.

## H    USE OF LARGE LANGUAGE MODELS (LLMs)

We used LLMs for editorial assistance (clarity, grammar, wording, and minor reorganizations). LLMs were not used to generate ideas, design experiments, or manipulate results, or draft related-work claims. The authors take full responsibility for all content.

