# OpenReview forum: "RPP: A Certified Poisoned-Sample Detection Framework for Backdoor Attacks under Dataset Imbalance"
_ICLR.cc/2026/Conference — Submitted to ICLR 2026_

### Official Review · Reviewer_oEwq · 2025-10-21

**Soundness:** 2
**Presentation:** 2
**Contribution:** 2
**Rating:** 2
**Confidence:** 5

**Summary:**

The authors consider the problem of bias due to class imbalance, particularly how it "amplifies" backdoor vulnerability. Experiments are conducted on image classifiers. The defense assumes an operational scenario (line 141,212) where backdoor triggers are to be detected among a pool (Z) of test samples. Based on the description of p. 5, I think the authors use a kind of training set "fuzzing" by random perturbations to try to discover backdoor triggers (poisoned samples) in a pool of data.

**Strengths:**

Addressing backdoors and bias is a challenging problem.

**Weaknesses:**

As the author ack on line 143, Observation 1 (line 161) has previously been noted.

The following reference, which adapts a backdoor defense to address model bias particularly to data imbalance, should have been cited particularly regarding Observation 2 (line 176):
[1] H. Wang et al.  Maximum Margin Based Activation Clipping for Post-Training Overfitting Mitigation in DNN Classifiers.  IEEE Trans. Artificial Intelligence (TAI), vol. 1, Aug. 2025.

How much data D is used for "Preliminary Model Training" (line 223) and where does it come from? I think it's the pool of test samples suggested in line 212. Under what practical scenarios does the pool necessarily contain backdoor triggers (poisoned samples) as assumed?

The description on lines 219- is confusing.

First, p(z|w) is poorly defined and it's inappropriate to cite another paper for a clear definition of such an important quantity to this paper.  In equation (3), p inherits a subscript which leads me to think it's a class posterior? How does equ (3) relate to equ (4) if at all? In any case, the notation in (3) and (4) is terrible.

Also, the authors appear to be working on a before/during training scenario where the possibly poisoned _training_ data (D) of the "preliminary" model  is available to the defense, in addition to a dataset V that is known to be benign with respect to a possible backdoor.  The following (uncited) work is also in this context but does not assume a benign dataset V is available to the defense:
Z. Xiang et al.  Reverse Engineering Imperceptible Backdoor Attacks on Deep Neural Networks for Detection and Training Set Cleansing.  Elsevier Computers & Security, 2021.
Also, before considering a possible threshold for Delta P, explain where sigma comes from.

The fonts in the figures are too small.

Overall, the paper's presentation is sloppy and relevant prior work is not adequately considered.

**Questions:**

See above.

---

> ### Author Response · Authors · 2025-11-21
>
> # Dear Reviewer oEwq,
> We sincerely thank you for your valuable time and constructive feedback. We'd like to address your concerns and provide clarifications as follows.
>
> (All modified contents are marked in blue in our revision.)
>
> > **Q1**: Observation 1(line 176) has been previously reported.
>
> **A1:**  Thank you for the comment. While prior work (line 147) noted that *long-tailed* distributions increase minority-class vulnerability, our work is the **first systematic analysis** of how different imbalance types—including *step imbalance*—impact backdoor vulnerability across diverse attack settings.
>
> > **Q2**: Cite [1]Maximum Margin-Based Activation Clipping for Post-Training Overfitting Mitigation in DNN Classifiers, for Observation 2.
>
> **A2:** Thanks for highlighting this work. We have added the following under Observation 2: Recent work (Wang et al., 2025) extends maximum-margin activation clipping (Wang et al., 2024b) from backdoor defense to mitigating model bias under imbalance. Although focused on post-training overfitting rather than poisoned-sample detection, its finding on margin distortion under imbalance supports our observation of degraded defense reliability.
>
> > **Q3**: How much data is used for “Preliminary Model Training,” and what is its source? What scenario would this pool contain poisoned samples?
>
> **A3:** Thanks for the question. A user collects a dataset 𝒟 (possibly imbalanced) from an untrusted source for downstream use. Before using, the user pretrains a classifier on 𝒟 to obtain probability vectors for all samples and a clean calibration set 𝒱. If 𝒟 contains poisoned data, the pretrained model will typically learn the backdoor (high ASR), allowing RPP to identify compromised samples. Similar assumptions are common in prior-training defenses: CT (Qi et al., 2023b) trains an initial model to assist detection, while SS (Tran et al., 2018) and AC (Chen et al., 2018) detect outliers by training directly on possibly poisoned data.
>
> > **Q4**: The description on lines 219(226)- is confusing.
>
> **A4:**  Revised version: Suppose 𝒟(x+δ) denotes the potentially poisoned dataset to be inspected, and 𝒱 represents a clean calibration set.
>
> > **Q5**: The definitions of p(z∣w), p inherits a subscript is a class posterior? Their relation in Eqs. (3)–(4) are unclear.
>
> **A5:**  Thank you for this comment. We'd like to clarify that $p(x∣w)$ has been defined in the *Formal Backdoor Attack Setting*(line 140): A classifier $f(\cdot|w)$ maps input $x$ to a predictive distribution $p(x|w)$. The $y$-th entry, $p_y(x|w)$, is defined as $p_y(x|w) = \mathbb{P}(f(x|w) = y)$.
>
> To improve clarity, we defined the **Samplewise Probability Vector (SPV)** (line 256):
>
> > For any $x$, The SPV is the predictive vector $p(x|w)$, where each component $p_y(x|w)=\mathbb{P}(f(x|w)=y)$ denotes the probability of assigning $x$ to class $y$. Under Gaussian noise ε∼N(0,σ²I), the noisy prediction is $p_y(x+ε|w)=\mathbb{P}(f(x+ε|w)=y)$.
>
> In Eq. (3), the subscript $y$ denotes the $y$-th entry of the predicted vector, following standard practice in randomized smoothing. $p(x)$ means $p_{y_t}(x+\delta|w)$ for a possibly poisoned input ($y_t$ is target class); Eq. (4) bounds its noisy counterpart $p_t=\mathbb{P}(f(x+\varepsilon|w)=y_t)\le\overline{p_t}$. Theorem 5.1 connects them to derive the certified lower bound on trigger size.
>
> > **Q6**: The authors assume a pre-training scenario with 𝒟 and clean set 𝒱. Related work[2] does not assume a clean 𝒱. [2]RE (Xiang et al., 2021).
>
> **A6:**  Thanks for highlighting this connection. RPP operates in a pre-training setting where the defender trains a preliminary model on possibly poisoned 𝒟 to obtain per-sample probabilities. Unlike empirical methods, RPP provides **certified detection guarantees**.
>
> While [2] also detects backdoors from the training data, it relies on heuristic reconstruction. And we have **compared RPP with 11 SOTA defenses under consistent settings.** Nevertheless, we appreciate reviewer’s suggestion and added a discussion in Related Work and included results in Table 2 and 11.
>
> #### Table. TPR/FPR for RPP vs. RE [2] on CIFAR-10 (ρ=2,100).
> Full results for ρ=1,10,200 are in Table 11.
> |ρ|Badnets|Blend|Trojan|SIG|ISSBA|WaNet|Sleeper Agent|AdaPatch|AdaBlend|Narci.|
> |-|-|-|-|-|-|-|-|-|-|-|
> |2|73.9/17.0|72.7/20.5|56.5/16.9|38.7/11.0|41.5/16.4|50.4/12.3|52.9/11.7|57.8/17.5|50.4/20.8|10.1/12.3|
> |100|13.6/22.5|10.4/14.3|20.2/11.0|14.4/15.9|16.4/15.9|17.0 /9.4|10.8/8.1|9.7/2.8|8.9/8.4|0.0/3.1|
>
> > **Q7**: Before defining a threshold for ΔP, explain where σ comes from.
>
> **A7:** Thanks for the comment. The noise parameter σ is defined in RPP(line 262) as the standard deviation of isotropic Gaussian perturbation ε∼N(0,σ²I). Similar to randomized smoothing(Cohen et al., 2019), σ controls perturbation magnitude and measures sample stability.
>
> > **Q8**: The fonts in the figures are too small.
>
> **A8:**  Thanks for the feedback. We will enlarge the fonts in the revised version.

---

### Official Review · Reviewer_tMra · 2025-10-28

**Soundness:** 3
**Presentation:** 3
**Contribution:** 2
**Rating:** 6
**Confidence:** 3

**Summary:**

The paper provides a well-motivated and well-supported certified defense framework addressing an important real-world issue in backdoor detection. Its contributions are technically solid and empirically convincing, though the conceptual novelty is somewhat limited.

**Strengths:**

The paper provides a thorough analysis of how dataset imbalance amplifies backdoor vulnerabilities and weakens existing defenses, which is a practical and underexplored aspect of backdoor research.

The work offers formal guarantees, including conditions for detectability, upper and lower trigger bounds, and provable control over false positive rate.

Evaluations on five benchmarks and ten attack types demonstrate strong performance, outperforming 11 state-of-the-art defenses in both detection accuracy and robustness under class imbalance.

**Weaknesses:**

The theoretical analysis assumes Gaussian perturbations and independent noise across samples, which may not hold for complex data distributions.

 Although adaptive scenarios are mentioned, the empirical defense-attack interplay is underexplored. More thorough experiments on adaptive trigger shaping or low-magnitude attacks would add credibility to claims of robustness.

**Questions:**

The theoretical analysis assumes Gaussian perturbations and independent noise across samples, which may not hold for complex data distributions.

 Although adaptive scenarios are mentioned, the empirical defense-attack interplay is underexplored. More thorough experiments on adaptive trigger shaping or low-magnitude attacks would add credibility to claims of robustness.

---

> ### Author Response · Authors · 2025-11-21
>
> # Dear Reviewer tMra,
>
> We sincerely thank you for your constructive feedback. We are encouraged by your positive comments on the **thorough analysis of dataset imbalance and backdoor vulnerability**, the **formal theoretical guarantees** for detectability and false-positive control, and the **strong empirical performance** across five benchmarks and ten attack types! We address your remaining concerns as follows.
>
> (All modified contents are marked in blue in our revision.)
>
> > **Q1**: The theoretical analysis assumes Gaussian perturbations and independent noise across samples, which may not hold for complex data distributions.
>
> **A1:**  We thank the reviewer for this insightful comment! Gaussian noise is commonly employed in certified robustness research due to its *analytical tractability* and *theoretical optimality* under the ℓ₂ norm. Its rotational symmetry and closed-form density facilitate rigorous derivation of robustness guarantees. Moreover, prior work has shown that among all isotropic noise distributions, Gaussian perturbations yield the tightest certification bound for ℓ₂-norm–bounded attacks[1-2].
>
> Accordingly, our theoretical analysis follows the standard *randomized smoothing framework* [1–2], which assumes i.i.d. Gaussian perturbations to ensure provable robustness guarantees. This assumption is consistently adopted in subsequent studies on certified robustness [3–6]. Empirically, we also observe that our theoretical findings remain valid even when real-world data deviate from this idealized Gaussian setting, as demonstrated by our experiments on MNIST, SVHN, CIFAR-10, TinyImageNet, and ImageNet.
>
> We have added a discussion in the *Limitations & Future Work* section(line 1643) acknowledging that our theoretical analysis relies on the assumption of i.i.d. Gaussian perturbations. Extending this analysis to encompass non-Gaussian or correlated perturbations could offer a more comprehensive understanding of robustness under realistic conditions and serves as a promising direction for future work.
>
> [1]Certified Adversarial Robustness via Randomized Smoothing. ICML, 2019.
>
> [2]Provably Robust Deep Learning via Adversarially Trained Smoothed Classifiers. NeurIPS, 2019.
>
> [3]Certified Robustness to Adversarial Examples with Differential Privacy. S&P, 2019.
>
> [4]Double Sampling Randomized Smoothing. PMLR, 2022.
>
> [5]Cbd: A certified backdoor detector based on local dominant probability. NeurIPS, 2024.
>
> [6]Bridging the Theoretical Gap in Randomized Smoothing. AISTATS, 2025.
>
> > **Q2**: Although adaptive scenarios are mentioned, the empirical defense-attack interplay is underexplored. More thorough experiments on adaptive trigger shaping or low-magnitude attacks would add credibility to claims of robustness.
>
> **A2:** Thank you for this insightful comment. We would like to clarify that our adaptive attacks already consider low-magnitude perturbations, where the adversary has full knowledge of our defense and deliberately chooses smaller triggers with $\lVert \delta \rVert_2 < 0.8$ (see line 479 and Appendix C.10). Because attack strength is characterized by the $\ell_2$ norm, we did not separately vary trigger shapes but instead controlled the overall perturbation magnitude to capture a broad range of stealthy scenarios.
>
> To further explore the empirical defense–attack interplay, we additionally evaluated a *hybrid-label adaptive attack*, in which the adversary, aware of RPP, poisons 80% of samples through standard label flipping and 20% with triggers mixed with Gaussian noise while preserving the original labels. This configuration emulates prediction instability in poisoned samples and aims to evade RPP detection.
>
> As shown in Table 1 (added in line 498 of the revised version), RPP maintains strong detection performance even under this adaptive setting.
>
> #### Table 1. RPP vs. Adaptive Attack across different imbalance ratios (ρ) with n = 100, α = 0.05 on CIFAR-10.
> | ρ       | TPR   | FPR   |
> |--------|-------|-------|
> | 1      | 90.0  | 6.8   |
> | 2      | 81.5  | 13.1  |
> | 10     | 79.0  | 12.5  |
> | 100    | 61.8  | 16.9  |
> | 200    | 59.7  | 26.4  |
>
> In all cases, ASR remained above 90% for detection.

---

### Official Review · Reviewer_ZaaY · 2025-10-31

**Soundness:** 3
**Presentation:** 3
**Contribution:** 3
**Rating:** 6
**Confidence:** 3

**Summary:**

This paper investigates the relationship between class imbalance and the vulnerability of deep neural networks to backdoor attacks. The authors empirically demonstrate that class imbalance in the training data can significantly increase the attack success rate (ASR), which is an important and insightful observation. To address this issue, the paper proposes a method for identifying poisoned samples caused by class imbalance. Unlike previous approaches that rely on global or class-level statistics (such as clustering or distribution estimation), the proposed method operates at the sample level, measuring the stability of output probabilities under random noise perturbations. Extensive experiments are conducted to validate its effectiveness.

**Strengths:**

1.The paper uncovers an important and previously underexplored phenomenon — that class imbalance can substantially amplify a model’s susceptibility to backdoor attacks. This observation provides a valuable perspective on how data distribution affects model security.

2.The proposed detection approach abandons conventional clustering- or distribution-based strategies, instead introducing a per-sample robustness criterion based on output stability under random perturbations. This idea is conceptually fresh, intuitive, and supported by convincing results.

3.The experiments cover various imbalance settings and provide strong evidence supporting both the motivation and the effectiveness of the proposed method.

**Weaknesses:**

1.Lack of validation on real-world long-tailed datasets:
All imbalance settings in the experiments are synthetically generated. The absence of evaluation on naturally imbalanced or long-tailed datasets  limits the practical generalizability of the results. Adding experiments on real datasets would significantly strengthen the paper’s claims.

2.Inconsistent notation:
Some symbols and notations are inconsistently used across sections, which slightly reduces readability. A careful revision to ensure consistency throughout the paper is recommended.

**Questions:**

1.How does the proposed approach perform on naturally imbalanced, long-tailed datasets instead of synthetic ones?

---

> ### Author Response · Authors · 2025-11-21
>
> # Dear Reviewer ZaaY,
>
> We sincerely thank you for your valuable time and constructive feedback. We appreciate your positive recognition of our work in revealing the impact of **class imbalance on backdoor vulnerability**, introducing a **novel per-sample robustness criterion**, and providing **comprehensive experimental validation**! We address your remaining concerns as follows.
>
> (All modified contents are marked in blue in our revision.)
>
> > **Q1**: How does the proposed approach perform on naturally imbalanced, long-tailed datasets instead of synthetic ones?
>
> **A1:** We thank the reviewer for this insightful question. As discussed in the *Limitations* section of the appendix(line 1637), our main experiments focus on synthetically imbalanced datasets. This setting is consistent with most existing studies on imbalanced learning, which primarily rely on artificially constructed long-tailed benchmarks for controlled evaluation [1-4].
>
> To further assess generalization in realistic settings, we additionally evaluated RPP on the naturally imbalanced iNaturalist-2018 dataset. As shown in Table 1 (added in the Appendix, line 1290), RPP maintains comparable detection performance across diverse attack types, demonstrating its robustness in real-world long-tailed scenarios.
>
> #### Table 1. RPP performance on iNaturalist-2018 under different backdoor attacks (n = 100, α = 0.05).
> | Attacks | TPR  | FPR  |
> |--------------|------|------|
> | Badnets      | 81.1 | 18.7 |
> | Blend        | 82.2 | 17.2 |
> | Trojan       | 74.8 | 15.6 |
> | SIG          | 77.0 | 10.9 |
> | ISSBA        | 78.5 | 24.3 |
> | WaNet        | 87.1 | 12.8 |
>
>
> [1]Long-Tailed Backdoor Attack Using Dynamic Data Augmentation Operations. arXiv, 2024.
>
> [2]Maximum Margin Based Activation Clipping for Post-Training Overfitting Mitigation in DNN Classifiers. TAI, 2025.
>
> [3]Decoupling Representation and Classifier  for Long-Tailed Recognition. CVPR, 2020.
>
> [4]Large-scale long-tailed recognition in an open world. CVPR, 2019.
>
> > **Q2**: Inconsistent notation: Some symbols and notations are inconsistently used across sections, which slightly reduces readability.
> >
>
> **A2:**  We thank the reviewer for this careful observation. We have carefully reviewed the entire manuscript to ensure that all mathematical symbols and notations are now used consistently across sections. In the revised version, all variables, indices, and probability notations follow a unified convention to improve clarity and readability.

---

### Official Review · Reviewer_YGmj · 2025-11-01

**Soundness:** 3
**Presentation:** 3
**Contribution:** 2
**Rating:** 4
**Confidence:** 3

**Summary:**

The paper firstly presents investigation of how the dataset imbalance amplifies backdoor vulnerability. To address this, the paper proposes Randomized Probability Perturbation(RPP) that operates in a black-box setting using only model output probabilities. RPP leverages sample-level prediction stability under noise perturbations and integrates conformal prediction to provide provable detection guarantees and control over the false positive rate. Extensive experiments confirm that RPP achieves higher detection accuracy, establishing a theoretically grounded framework for defending against backdoor attacks with imbalanced data.

**Strengths:**

1. Originality:

   The paper presents the first work to systematically investigate and formalize the critical problem of how dataset imbalance amplifies backdoor vulnerability and cripples existing defenses.

   The paper proposes Randomized Probability Perturbation(RPP), which adapts the concept of randomized smoothing to the distinct task of certified poisoned sample detection, proposing a paradigm shift from distribution-level signals used by prior defenses to a sample-level robustness metric.

2. Quality:

   The paper provides strong theoretical foundations, with formal guarantees for certified detectability (Theorem 5.1) and False Positive Rate control (Theorems 5.3, 5.4).

   The experimental section encompasses an extensive benchmark: 5 datasets, 2 imbalance types, 4 imbalance ratios, 10 diverse backdoor attacks, and 11 state-of-the-art baseline defenses. The results are consistent and support the paper's claims.

3. Clarity:

   The motivation is established in the introduction, which effectively critiques prior work and outlines the paper's contributions. The logical flow from problem identification to solution and evaluation is smooth and easy to follow.

4. Significance:

   RPP provides a black-box and certified tool for securing ML pipelines in critical domains like medical diagnosis and autonomous driving, where data is inherently imbalanced and security is paramount.

**Weaknesses:**

1. The paper's sample-level approach is commendable, but its global calibration of RPP thresholds remains vulnerable to distributional skew from class imbalance. A critical analysis of per-class detection performance and adaptive thresholding strategies is needed.
2. The practicality of requiring a pre-trained preliminary model on potentially poisoned data needs clearer justification regarding security and computational overhead. The exact deployment scenario and cost-benefit analysis compared to end-to-end defenses should be elaborated.

**Questions:**

1. Your method uses a global threshold derived from the calibration set. Given that the model's confidence and behavior can vary significantly between majority and minority classes, have you observed any systematic difference in the separation of clean vs. poisoned RPP values across different classes? Does the detection performance for poisoned samples degrade in the tail classes compared to the head classes?
2. Could you please elaborate on the precise real-world scenario for deploying this "prior-training" defense? Specifically, how does a defender obtain a "preliminary model" that has learned the backdoor without already being compromised?
3. The core intuition is that poisoned samples have lower RPP. How does the full RPP method (with conformal prediction) compare against a much simpler baseline that directly thresholds the empirical  $\tilde{\Delta}P$ value, or against a baseline that flags a sample if its top-1 prediction is invariant over $J$ noise perturbations?

---

> ### Author Response · Authors · 2025-11-21
>
> # Dear Reviewer YGmj,
>
> We sincerely thank you for your valuable time and constructive feedback. We appreciate your positive comments on the **originality, theoretical soundness, comprehensive experiments**, and **practical significance** of RPP for certified detection under data imbalance! We address your remaining concerns as follows.
>
> (All modified contents are marked in blue in our revision.)
>
> > **Q1**: The global calibration of RPP thresholds may be affected by class imbalance. Any differences in clean vs. poisoned RPP separation or degradation in tail-class detection.
>
> **A1:** Thank you for the insightful comment. While the calibration threshold of RPP is global, it's derived via **conformal prediction**, which adaptively calibrates thresholds in a *distribution-adaptive* manner using clean calibration data, maintaining reliability under imbalance.
>
> We observed imbalanced datasets are more vulnerable to backdoors. Our threat model assumes the attacker exploits imbalance by poisoning minority-class samples and relabels them as majority class(0) to reach high ASR with minimal budget. Thus, poisoned data exist only in the majority class. Shown in Table 1, minority classes(1–9) exhibit low FPR, consistent with this setting. TPR for them is omitted since they contain no poisoned samples under the many-to-one scenario.
>
> We do not consider multi-target backdoors[1], as even empirical detection remains challenging[2].
>
> #### Table 1. Per-class and global TPR/FPR of RPP(varying ρ).
> |ρ|Global TPR|Global FPR|0 TPR|0 FPR|1 FPR|2 FPR|3 FPR|4 FPR|5 FPR|6 FPR|7 FPR|8 FPR|9 FPR|
> |-|:-:|:-:|:-:|:-:|:-:|:-:|:-:|:-:|:-:|:-:|:-:|:-:|:-:|
> |1|98.5|5.9|98.5|8.7|5.0|5.7|5.8|5.6|4.6|5.4|6.0|5.5|6.7|
> |10|96.7|3.2|96.7|6.1|0.6|0.0|0.0|0.0|0.9|0.8|0.0|0.0|1.3|
> |200|83.3|14.3|83.3|14.4|0.0|0.3|0.2|0.0|0.1|0.0|0.1|0.1|0.6|
>
> [1]One-to-n & n-to-one: Two advanced backdoor attacks against deep learning models. TDSC.
> [2]Cbd: A certified backdoor detector based on local dominant probability. NeurIPS.
>
> >**Q2**: Clarify the practicality of the prior-training stage—how the “preliminary model” is obtained without being compromised and its efficiency vs. end-to-end defenses.
>
> **A2:**  Thank you for this important question! A user collects a dataset 𝒟 (possibly imbalanced) from an untrusted source for on-device or downstream use. Before deployment, the user pretrains a lightweight preliminary model on 𝒟 to obtain predictive probability vectors.
>
> If 𝒟 contains poisoned data, this model typically learns the backdoor (high ASR), enabling RPP to detect compromised samples via probability perturbation. Crucially, this model is *not deployed or reused*; it serves only as an **intermediate artifact** for certified detection.
>
> Similar assumptions are widely adopted in prior training defenses. CT[1] trains an initial model to aid detection; SS[2] and AC[3] train on possibly poisoned data and flag outliers in the learned space.
>
> The prior-training stage is lightweight (few epochs) and requires no model modification. In contrast, end-to-end defenses such as Fine-Pruning or NAD require heavy retraining costs. Thus, RPP achieves *certified poisoned-sample detection* with minimal overhead and no dependency on model internals.
>
> [1]Towards a proactive {ML} approach for detecting backdoor poison samples. USENIX.
> [2]Spectral signatures in backdoor attacks. NeurIPS.
> [3]Detecting backdoor attacks on deep neural networks by activation clustering. arXiv.
>
> > **Q3**: The core intuition is that poisoned samples have lower RPP. How does full RPP (with conformal prediction) compare to simpler baselines that directly threshold $\tilde{\Delta}P$ or flag invariant top-1 predictions under J perturbations?
>
> **A3:**  We appreciate the suggestion. A naïve baseline that thresholds $\tilde{\Delta}P$ or flags invariant top-1 predictions can partly reflect RPP’s idea, but such heuristics lack **statistical calibration**, making FPR uncontrollable.
>
> RPP employs conformal prediction, which adaptively calibrates thresholds using clean calibration data, ensuring robustness under imbalance. In Table 2 (line 443), fixed-threshold baselines like SS, SCALE-UP, and BBCaL degrade under imbalance.
>
> We also implemented a Top-1 invariance baseline, perturbing each sample J=3 times with Gaussian noise (σ=1.0). A sample is flagged if its top-1 label remains unchanged. Although it achieves high TPR, it yields very high FPR(Table 1)—especially for imbalanced data—since clean samples often preserve semantics and stable predictions.
>
> RPP instead measures the $\ell_\infty$-norm difference between full prediction vectors of clean and noisy samples, capturing subtle probability shifts even when the top-1 label stays constant.
>
> #### Table 1. Top-1 Invariance on CIFAR-10 (Badnets/Narci., varying ρ)
> |ρ|Badnets TPR|Badnets FPR|Narci. TPR|Narci. FPR|
> |:-:|:-:|:-:|:-:|:-:|
> |1|95.4|40.4|96.1|45.8|
> |2|95.9|53.4|97.3|51.2|
> |10|98.0|78.5|99.9|68.3|
> |100|100.0|97.7|99.7|97.9|
> |200|100.0|96.1|100.0|99.6|

---

### Comment · Area_Chair_qwkV · 2025-11-24

Please respond to the authors' rebuttal. Thanks.

AC

---

### Meta-Review · Area_Chair_Jtkg · 2026-01-06

**Summary:**

The paper systematically investigates how class imbalance in training datasets amplifies a model's vulnerability to backdoor attacks and undermines existing defenses. To address this, the authors propose Randomized Probability Perturbation (RPP), a sample-level detection framework that operates in a black-box setting. Unlike prior defenses that rely on distribution-level statistics, RPP utilizes randomized smoothing concepts to measure per-sample prediction stability under noise. The framework incorporates conformal prediction to provide formal theoretical guarantees for certified detectability and False Positive Rate (FPR) control. The method was evaluated across five datasets and ten diverse attack types, demonstrating superior performance over 11 state-of-the-art baselines.

**Reviewer Concerns:**

The review process revealed several significant concerns regarding the practicality and empirical scope of the work. A major point of contention (Reviewers YGmj, oEwq) is the requirement for a "preliminary model" trained on potentially poisoned data; reviewers questioned the realism of a scenario where a defender possesses such a model without it already being compromised. Additionally, Reviewer ZaaY noted that the experiments relied exclusively on synthetically generated imbalance, lacking validation on real-world long-tailed datasets. Reviewer oEwq raised serious concerns about the presentation quality, citing inconsistent mathematical notation (e.g., in Equations 3 and 4) and a failure to cite relevant prior work regarding model bias and backdoor defense (e.g., Wang et al., 2025). Finally, the sensitivity of the global threshold to class-specific distributional skew was highlighted as a potential weakness in highly imbalanced settings.

**Reviewer Scores:**

The scores are highly polarized at 6, 6, 4, and 2, reflecting a lack of consensus. While Reviewers ZaaY and tMra (6, 6) praised the "conceptually fresh" sample-level approach and the solid theoretical grounding, Reviewers YGmj (4) and oEwq (2) remained skeptical. Despite the authors' rebuttal, the reviewers chose not to upgrade their scores, suggesting that the clarifications on the "prior-training" deployment scenario and the notation issues did not fully satisfy their concerns. Specifically, the harsh "Reject" from Reviewer oEwq, based on perceived lack of novelty and poor clarity, significantly drags down the average. Given the remaining doubts about the method's practical generalizability and the presentation flaws that were not resolved to the reviewers' satisfaction, the Area Chair recommends Rejection.

---

### Decision · Program_Chairs · 2026-01-26

Reject